# Spatial heterogeneity of the nonlinear impact of built environment on suburban railway commuting willingness: Taking Guiyang Loop Line Railway as an example

Rufeng Chen , Hang Zhao *

School of Geography & Environmental Science, Guizhou Normal University, Guiyang, Guizhou, China

* wazua@163.com

## Abstract

This study takes the Guiyang Loop Line Railway as a case study and, based on resident commuting survey data, applies an XGBoost model combined with Individual Conditional Expectation (ICE) analysis to examine the threshold effects and non-linear spatial characteristics of the built environment on residents' willingness to commute via the Loop Line Railway. The results indicate that: (1) population density, distance to the nearest bus stop, road network density, and distance to the city center are key built environment factors influencing commuting willingness. Among them, population density and road network density exhibit overall positive effects, whereas distance to the nearest bus stop and distance to the city center show negative effects; (2) the spatial distributions of the threshold effect magnitude (Δy) and the non-linear effect intensity (m) display a heterogeneous pattern broadly characterized by a "city center–suburban–peripheral" gradient, suggesting pronounced spatial non-stationarity in the relationship between the built environment and commuting willingness; (3) compared with the central urban area, residents living around suburban and peripheral stations are more sensitive to changes in bus feeder conditions, road network connectivity, and commuting distance, exhibiting stronger threshold effects and non-linear responses. These findings suggest that enhancing residents' willingness to commute via the Loop Line Railway requires context-sensitive strategies that account for the built environment characteristics of different areas, with targeted improvements in bus–rail integration and road network conditions, rather than adopting uniform development and supporting policies across all stations.

## Introduction

Suburban railways, often termed commuter rail or suburban rail, are categorized as a form of passenger rail system positioned between urban rail transit (e.g., metro systems) and intercity railways; they are characterized by commuter-oriented services,

**Data availability statement:** All data files are available from the figshare database (https://doi.org/10.6084/m9.figshare.31146256).

**Funding:** The research is supported by the National Natural Science Foundation of China [Grant No. 71864008].The funder website is https://www.nsfc.gov.cn/. The funder did not participate in the research design, data collection and analysis, publication decisions, or manuscript writing.

**Competing interests:** The authors have declared that no competing interests exist.

relatively high operating speeds, and large transport capacity [1]. As a long-distance commuting mode, a crucial role is played by suburban railways in connecting central urban areas with peripheral suburbs and in supporting the development of a "one-hour commuting circle". In comparison to conventional urban rail transit, a wider spatial area is typically served by suburban railways, which feature larger station spacing, higher operating speeds, and lower service frequencies. They are primarily designed to accommodate medium-to long-distance commuting between central cities and surrounding suburban areas or satellite towns. Their functional orientation is focused on establishing efficient commuting corridors within metropolitan areas [2–3]. In this context, a significant contribution is made by suburban railways toward relieving functional pressure in central urban areas, shaping rational commuting structures, facilitating convenient daily mobility, and improving accessibility in peripheral urban regions [4].

With the continuous expansion of urban space and the deepening spatial mismatch between residential and employment locations, increasing scholarly attention has been attracted by suburban railways worldwide. The potential role of suburban rail systems in supporting metropolitan commuting structures has been highlighted by existing studies, viewed from perspectives such as urban land-use optimization, spatial structure refinement, and regional integration [5–6]. However, within the Chinese context, intended commuting functions have not been fully realized by many operational suburban railway lines, which commonly face problems of low ridership and underutilization [7–8]. To explain these phenomena, the limited commuting performance of suburban railways has been mainly attributed by previous studies to operational or institutional factors, including inappropriate station location, relatively high fares, long waiting times, and insufficient supporting infrastructure [9–11].

However, for suburban railways characterized by larger station spacing and broader service catchments, the conversion of commuting willingness into actual ridership is not only dependent on train operating services but is also strongly constrained by the built environment conditions surrounding residents' places of dwelling. In contrast to urban metro systems, a higher sensitivity to first- and last-mile travel conditions is exhibited by commuting behavior on suburban railways. When feeder bus services are insufficient, road network connectivity is poor, or station accessibility is low, the adoption of suburban railways as a regular commuting mode may be difficult for residents, even when latent commuting willingness exists. Consequently, persistently low utilization of such lines is often observed. Nevertheless, operational and service supply factors have been the primary focus of existing studies on insufficient ridership, while limited attention has been paid to the mechanisms through which commuting willingness is influenced by the built environment, ultimately leading to low passenger volumes. On the one hand, a station-area (destination-end) perspective has been primarily adopted in prior research, emphasizing land use patterns or development intensity around stations; conversely, systematic empirical evidence regarding the effect of residential-end built environment characteristics on commuters' mode choice remains scarce [12]. On the other hand, most existing studies are based on large cities with relatively simple topographic conditions [13]. Mountainous

cities, where urban spatial structure is strongly constrained by natural terrain, have received far less attention, and the applicability of existing findings to such contexts has yet to be adequately examined.

To address these research gaps, this study investigates how residential built environment characteristics influence commuters' willingness to use suburban railways, using Guiyang, China—a typical mountainous city—as an empirical case. Specifically, this study aims to: (1) examine the nonlinear relationships between residential built environment factors and suburban rail commuting willingness; (2) identify threshold effects and spatial heterogeneity of key influencing variables using machine learning approaches; and (3) provide planning implications for improving suburban railway utilization in mountainous cities. This study contributes to the existing literature in three aspects. First, it shifts the analytical perspective from station-area characteristics to residential-side built environment factors influencing suburban rail commuting behavior. Second, it extends suburban rail research to mountainous urban contexts where terrain constraints play an important role. Third, it applies data-driven methods to reveal nonlinear and heterogeneous mechanisms underlying suburban rail use, thereby providing new insights into improving suburban railway performance.

## Literature review

### Behavioral theories and commuting intention

In travel behavior research, behavioral theories have been widely applied to explain individuals' travel intentions and mode choice decisions. Among them, the Theory of Planned Behavior (TPB) and the Technology Acceptance Model (TAM) are two of the most commonly adopted theoretical frameworks. According to the Theory of Planned Behavior proposed by Ajzen, individuals' behavioral intentions are jointly determined by attitude, subjective norms, and perceived behavioral control [14]. This framework has been extensively applied in transportation studies to explain travel mode choice behavior. Empirical evidence suggests that public transport use is significantly influenced by these three components, with attitude often identified as the most influential factor shaping travel decisions [15–17]. Other studies further highlight the critical roles of attitude and perceived behavioral control in determining individuals' intentions to use public transportation [18–19]. The Technology Acceptance Model (TAM), originally developed by Davis [20], emphasizes perceived usefulness and perceived ease of use as key determinants influencing individuals' acceptance of new technologies. With the emergence of new transportation modes, TAM has increasingly been applied in transportation research, including studies on electric vehicle adoption and new energy bicycle usage [21–22]. To provide more comprehensive behavioral explanations, some studies have integrated TPB and TAM frameworks. For example, Chen combined TPB, TAM, and habit theory to analyze private car users' willingness to shift to public transport [23]. Similarly, Wang employed an integrated TPB–TAM framework and found that perceived usefulness, perceived ease of use, attitude, subjective norms, and perceived behavioral control jointly influence individuals' willingness to adopt sustainable travel modes [24]. These studies provide an important theoretical foundation for examining commuting willingness in emerging transportation contexts.

### Suburban railways and commuting research

Commuting is defined as the long-term, regular, and repetitive round-trip travel between individuals' places of residence and employment, characterized by high regularity, persistence, and temporal rigidity [25]. With rapid economic development and continuous urban expansion, the spatial separation between workplaces and residences has increased significantly, resulting in longer commuting distances and travel times. These changes contribute to traffic congestion, reduced travel efficiency, and decreased commuter satisfaction, thereby constraining sustainable urban development [26]. Due to their high operating speed, large passenger capacity, and long service distance, suburban railways are widely regarded as critical infrastructure for supporting medium- and long-distance commuting and for facilitating "one-hour commuting circles" in metropolitan regions [27]. In Europe, North America, and East Asia, suburban rail systems were developed relatively early and have achieved relatively mature operational frameworks, as exemplified by the Paris suburban railway

network [28], the Tokyo commuter rail system [29], and commuter rail systems in the United States [30]. Existing studies in these contexts primarily focus on commuting demand structures, passenger flow characteristics, and operational efficiency, providing valuable insights into the role of suburban railways in metropolitan commuting systems.

In contrast, suburban railways in China remain in an exploratory and developmental stage. Although several lines, such as the Shanghai R Line [31] and the Beijing S Line [32], have been commissioned, commuting ridership performance has generally been unsatisfactory, and intended commuting functions have not been fully realized. Previous studies have largely attributed this outcome to factors related to commuter behavior and operational service attributes. For instance, limitations in route alignment, pricing, and operational flexibility were identified by Alimo as major contributors to low suburban rail ridership [11]. Irawan found that walking distance, waiting time, and parking costs significantly influence commuters' mode choice decisions [33]. Similar conclusions have been drawn in studies conducted in China, which highlight service frequency, travel time [9], travel cost [34], infrastructure conditions, and development strategies [10] as key determinants of suburban rail usage.

Beyond operational attributes and fare policies, recent research has begun to emphasize the importance of commuting trip chains and access–egress conditions. Brohi, drawing on the Theory of Planned Behavior, demonstrated that commuters' attitudes toward public transport significantly influence their willingness to use suburban railways [35]. Le further suggested that maintaining affordable fares while improving last-mile services—such as bike-sharing systems and feeder bus connections—can substantially enhance the commuting attractiveness of suburban railways [36]. Overall, although existing studies have provided a relatively comprehensive understanding of suburban railway commuting from behavioral and operational perspectives, most research remains focused on supply-side explanations or treats entire railway lines as the unit of analysis.

## Built environment effects and research gaps

Against this backdrop, increasing attention has been paid to the influence of the built environment on residents' commuting mode choice. Built environment characteristics are commonly operationalized using the well-established "5D" framework—density, diversity, destination accessibility, design, and distance to transit—to evaluate their effects on travel behavior [37]. For example, Jia demonstrated that built environment attributes such as block size and intersection density are strongly associated with residents' mode choice decisions [38]. Wang examined the driving mechanisms of urban rail transit ridership and revealed non-linear relationships and interaction effects between built environment factors and ridership, thereby providing empirical support for rail transit planning [39]. Similarly, Xi identified spatiotemporal non-linear relationships between metro station built environments and passenger flows, showing that factors such as population density and parking facility density exert significantly non-linear impacts on station-level ridership [40].

Although extensive research has confirmed that built environment attributes—such as population density and land-use mix—affect commuting mode choice by shaping travel distance, accessibility, and transfer convenience [41], most existing studies focus on urban rail transit and metro systems. Empirical evidence on suburban railways remains limited. Compared with metro systems, suburban railways typically have larger station spacing and broader service catchment areas, placing greater emphasis on access and egress conditions. As a result, commuters may be more sensitive to variations in the surrounding built environment in suburban railway contexts [42]. This further complicates the identification of non-linear behavioral responses and spatial differences in commuting willingness across different urban locations.

While a small number of recent studies have begun to examine non-linear built environment effects on ridership and travel willingness, systematic analyses of spatial heterogeneity across stations within the same suburban railway line remain scarce. Therefore, a clear research gap exists regarding how residential built environment characteristics influence suburban railway commuting willingness, particularly in terms of non-linear effects and spatial heterogeneity. Addressing this gap constitutes the core objective of the present study.

## Research design

### Study area

Guiyang, the capital city of Guizhou Province in southwest China, is selected as the study area. As a typical mountainous city, Guiyang is characterized by complex topography and significant elevation differences, which impose substantial constraints on urban spatial expansion and transport infrastructure development. As of October 2024, the density of the urban road network was approximately 6.7 km/km², lower than the national recommended standard of 8 km/km² [43]. Meanwhile, the number of registered motor vehicles has been increasing by approximately 20,000 per month, indicating rapid growth in travel demand under relatively limited road supply conditions. These characteristics create distinctive commuting patterns and transportation challenges compared with cities located in flat terrains. The Guiyang Loop Line Railway, China's first urban loop suburban railway, was officially opened on March 30, 2022. The line has a total length of approximately 113 km with 17 stations, forming an important transportation corridor connecting multiple residential districts and urban functional areas and supporting the development of a "one-hour commuting circle" [44]. Despite its strategic positioning, the line has experienced relatively low ridership levels since its operation, with an average daily passenger volume of approximately 3,000 and high seat vacancy rates during certain periods [3]. The coexistence of substantial commuting demand and relatively low rail utilization makes Guiyang an appropriate case for examining the relationship between residential built environment characteristics and suburban rail commuting behavior in terrain-constrained cities.

### Sample selection and survey site determination

This study investigates residents' willingness to commute via the Guiyang Loop Line Railway. Given that suburban railways primarily facilitate medium- to long-distance commuting between urban cores and peripheral areas, they are characterized by large inter-station spacing and significant reliance on access and egress connections. Consequently, conventional catchment area delineations based on walking distances, which are commonly applied to urban metro systems, are not directly applicable in this context. Previous studies indicate that users of suburban or commuter rail systems rely heavily on cycling, feeder buses, and other modes—in addition to walking—to complete first- and last-mile trips; as a result, acceptable access distances substantially exceed the typical walking thresholds observed for metro systems [45]. Further empirical evidence suggests that commuters' tolerance regarding access and egress is constrained more significantly by temporal thresholds than by physical distance alone [46]. Field investigations conducted for this study indicate that, in commuting contexts, passengers generally accept access and egress travel times of 15–20 minutes, which corresponds approximately to a spatial radius of 2–3 km. In comparison to larger catchment definitions, a 3 km buffer effectively captures the primary residential clusters surrounding stations while excluding distant samples that exhibit weak relevance to suburban rail commuting decisions. Therefore, considering the functional characteristics of suburban railways, station spacing, and commuters' access behavior, the research area in this study is defined as a 3 km radius around each station.

The survey encompassed all operational stations along the Guiyang Loop Line Railway. To ensure spatial representativeness, questionnaire allocation was proportional to the population size within each station's 3 km buffer: stations characterized by higher population density and greater functional centrality were assigned larger sample sizes, whereas peripheral suburban stations were allocated fewer questionnaires. The specific survey locations and study areas are illustrated in Fig 1.

The questionnaire design was centered on the mechanisms underlying residents' willingness to commute via the Loop Line Railway. Data regarding respondents' commuting characteristics, alongside subjective perceptions of service performance, station conditions, and associated railway attributes, were collected. Prior to the full-scale survey, a pilot study was conducted to refine and simplify the wording, thereby minimizing ambiguity and enhancing response quality. The primary survey was administered between April and May 2024 via face-to-face paper-based questionnaires distributed within a 3 km radius of each station. In total, 750 questionnaires were distributed. Following the exclusion of questionnaires with

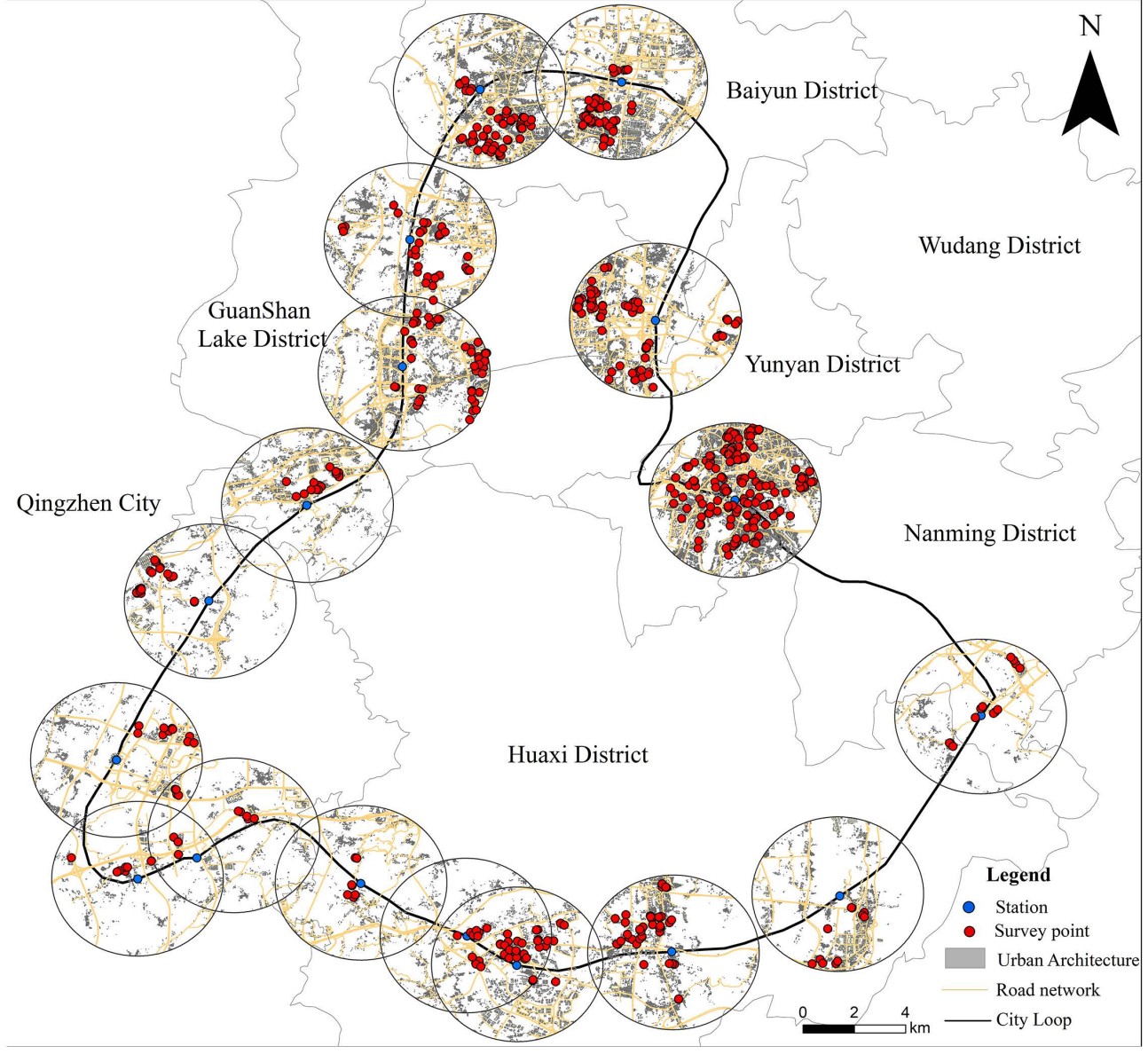

**Fig 1. Schematic Diagram of Survey Points and Study Area.** Note: This figure was produced by the author using ArcGIS.

incomplete responses or significant logical inconsistencies, 700 valid questionnaires were retained. During the model estimation phase, a secondary screening of the sample was performed to ensure the analysis adequately captured the differential effects of the built environment on commuting willingness across varied spatial contexts. Specifically, a subset of 451 questionnaires was selected for analysis, drawn from six representative stations: Guiyang Station, Guiyang North Station, Baiyun North Station, Jinyang South Station, Gui'an Station, and Huaxi South Station. Substantial variations in built environment characteristics, spatial location, infrastructure provision, functional roles, and ridership levels are exhibited by these stations. Collectively, they represent core transportation hubs, regional sub-centers, and standard functional stations, reflecting a spatial gradient spanning from the urban center to suburban and peripheral zones. This sampling

strategy facilitates a comprehensive examination of the heterogeneous effects of the built environment on individuals' willingness to commute via the Loop Line Railway across diverse urban contexts.

## Theoretical foundations and questionnaire design

Individual commuting mode choice is influenced not only by objective built environment conditions but is also strongly contingent upon individuals' subjective perceptions of travel convenience, accessibility, and service quality. The Theory of Planned Behavior (TPB) [14] and the Technology Acceptance Model (TAM) [20] are recognized as widely applied frameworks for explaining travel mode choice behavior. Consequently, these two theories are adopted in this study as the theoretical foundation for analyzing residents' commuting behavior. In TPB, it is emphasized that individuals' behavioral intention is jointly determined by attitudes, subjective norms, and perceived behavioral control [47]. The focus of TAM lies in individuals' subjective evaluations of the perceived usefulness and perceived ease of use of transport systems. This model is considered particularly suitable for explaining acceptance behavior toward emerging transport modes [48]. As shown in Fig 2.

Building on these frameworks—and given the high dependence of suburban railway commuting on station-area environments and service experiences [49]—satisfaction with the Loop Line Railway, perceived station conditions, and perceived road conditions are introduced as contextual extension variables. These variables are incorporated to address the limitations of traditional behavioral theories in capturing spatial and infrastructure-related factors. Variable definitions and measurements are summarized in Table 1.

The questionnaire comprises nine dimensions and 32 measurement items. Perceived usefulness measures residents' evaluations of the Loop Line Railway in terms of commuting efficiency, economic benefits, and improvements in physical and mental well-being. Perceived ease of use reflects residents' perceptions of convenience in accessing stations, obtaining operational information, and purchasing tickets. Subjective norms capture the influence of online publicity, fare policies, and attitudes of family members and friends on commuting choices. Satisfaction with the Loop Line Railway evaluates timetable arrangements for commuting, riding comfort, and service quality. Station-aware and road condition perception reflect the effects of station layout, feeder facilities, and access road characteristics on commuting willingness. Perceived behavioral control assesses residents' judgments regarding their ability to use the Loop Line Railway and tolerate relatively complex commuting procedures. Attitude represents residents' overall evaluations of the convenience, usefulness, and comfort of commuting by the Loop Line Railway. Finally, commuting willingness reflects residents' intentions to choose the Loop Line Railway for commuting under current conditions and to recommend it to others.

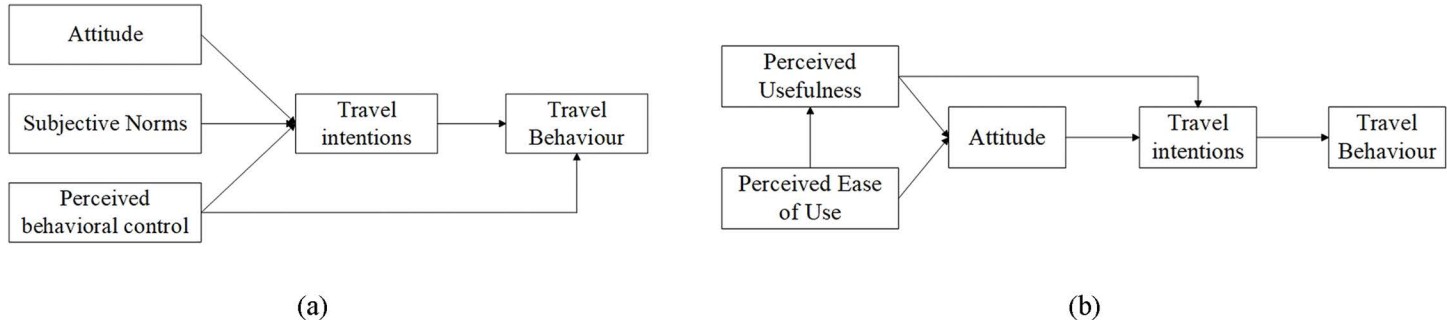

(a)  (b)

**Fig 2. Foundational Model of Travel Behaviour.** Note: A the Theory of Planned Behavior; b the Technology Acceptance Model. This figure was produced by the author using Visio.

**Table 1. Commuter willingness survey questionnaire for the Loop Line Railway.**

| Measurement dimension | Measurement Item |
|---|---|
| **Perceived usefulness** | Commuting efficiency |
| | Economic benefits |
| | Well-being improvement |
| **Perceived ease of use** | Access convenience |
| | Learning ease |
| | Information accessibility |
| | Ticketing convenience |
| **Subjective norms** | Online publicity impact |
| | Fare elasticity |
| | Peer usage |
| | Social imitation |
| | Social encouragement |
| **Satisfaction with the Loop Line Railway** | Schedule suitability |
| | Comfortable travel environment |
| | Reasonable commuter fares |
| | Service attitude |
| | Operation frequency |
| **Station-aware** | Station layout |
| | Feeder facilities |
| | Land-use development |
| **Road condition perception** | Road gradient |
| | Road circuity |
| | Road width |
| **Perceived behavioral control** | Behavioral ease |
| | App proficiency |
| | Procedural resilience |
| **Attitude** | Commuting convenience |
| | Practicality |
| | Perceived comfort |
| **Commuting willingness** | Operational satisfaction |
| | Modal preference |
| | Recommendation intention |

## Variable definitions and descriptive statistics

Residents' willingness to commute via the Guiyang Loop Line Railway serves as the dependent variable, measured using a seven-point Likert scale where higher values indicate stronger willingness. Descriptive statistics of the dependent variable are presented in Table 2. Drawing upon the "5D" built environment framework, 15 built environment indicators within an 800-m buffer of residences are employed to examine the mechanisms through which commuting willingness is influenced by residential built environment characteristics (see Table 3). Specifically, population density is used to capture the level of population concentration in residential areas; distance to the nearest bus stop reflects the convenience of access to feeder public transport; road network density represents the supply level of road

**Table 2. Descriptive characteristics of latent variable scores.**

| Latent Variable | Observed Variable | Maximum | Minimum | Mean | Overall Mean |
|---|---|---|---|---|---|
| **Perceived usefulness** | PU1 | 7 | 1 | 4.80 | 4.73 |
| | PU2 | 7 | 1 | 4.64 | |
| | PU3 | 7 | 1 | 4.75 | |
| **Perceived ease of use** | PEU1 | 7 | 1 | 4.00 | 4.44 |
| | PEU2 | 7 | 1 | 4.03 | |
| | PEU3 | 7 | 1 | 4.70 | |
| | PEU4 | 7 | 1 | 5.04 | |
| **Subjective norms** | SN1 | 7 | 1 | 4.41 | 4.29 |
| | SN2 | 7 | 1 | 4.56 | |
| | SN3 | 7 | 1 | 3.92 | |
| | SN4 | 7 | 1 | 4.25 | |
| | SN5 | 7 | 1 | 4.32 | |
| **Perceived behavioral control** | PBC1 | 7 | 1 | 4.42 | 4.44 |
| | PBC2 | 7 | 1 | 4.65 | |
| | PBC3 | 7 | 1 | 4.24 | |
| **Attitude** | AT1 | 7 | 1 | 3. 84 | 4.10 |
| | AT2 | 7 | 1 | 4. 17 | |
| | AT3 | 7 | 1 | 4.28 | |
| **Satisfaction with the Loop Line Railway** | RS1 | 7 | 1 | 4.74 | 4.65 |
| | RS2 | 7 | 1 | 4.96 | |
| | RS3 | 7 | 1 | 4.67 | |
| | RS4 | 7 | 1 | 4.87 | |
| | RS5 | 7 | 1 | 4.01 | |
| **Station-aware** | SA1 | 7 | 1 | 4.65 | 4.64 |
| | SA2 | 7 | 1 | 4.68 | |
| | SA3 | 7 | 1 | 4.60 | |
| **Road condition perception** | RCA1 | 7 | 1 | 4.06 | 4.23 |
| | RCA2 | 7 | 1 | 4.36 | |
| | RCA3 | 7 | 1 | 4.26 | |
| **Commuting willingness** | CW1 | 7 | 1 | 3. 93 | 4.34 |
| | CW2 | 7 | 1 | 4.83 | |
| | CW3 | 7 | 1 | 4.25 | |

infrastructure surrounding the residence; and distance to the city center characterizes the spatial location and centrality of the residential area.

## XGBoost model

Extreme Gradient Boosting (XGBoost) is an ensemble learning algorithm based on gradient boosting. It enhances predictive performance and generalisation capabilities by combining a series of decision tree models [50]. Given a training dataset $D = \{(x_i, y_i)\}_{i=1}^{n}$ the mode's predicted value is expressed as:

$$\hat{y}_i = \sum_{k=1}^{K} f_k(x_i), f_k \in F$$

(1)

**Table 3. Descriptive statistics of built environment indicators.**

| Dimension | Variable | Minimum | Maximum | Mean |
|---|---|---|---|---|
| Density | Population Density (10,000 persons/km²) | 0.02 | 4.79 | 2.15 |
| Design | Road Network Density/(km/km²) | 2.39 | 19.22 | 9.99 |
| Diversity | Density of public facilities (units/km²) | 0.00 | 28.35 | 5.81 |
| | Company density (per km²) | 3.48 | 363.13 | 80.93 |
| | Commercial residential density (units/km²) | 0.99 | 536.74 | 90.26 |
| | Automotive service density (units/km²) | 0.50 | 132.82 | 26.08 |
| | Healthcare density (units/km²) | 1.49 | 153.71 | 49.97 |
| | Shopping density (units/km²) | 0.50 | 1296.33 | 291.01 |
| | Restaurant density (units/km²) | 0.00 | 983.94 | 186.86 |
| Bus Proximity | Distance to Nearest Bus Stop (km) | 0.03 | 0.68 | 0.21 |
| | Bus Stop Density (unit/km²) | 0.50 | 16.42 | 5.60 |
| | Bus Line Density (per/km²) | 0.50 | 77.10 | 30.05 |
| | Distance to Nearest Metro Station (km) | 0.07 | 17.91 | 1.63 |
| Accessibility | Distance from City Center (km) | 0.12 | 23.71 | 7.81 |
| Newly added | Distance to Nearest Loop Line Railway Station (km) | 0.22 | 3.00 | 1.76 |

In Equation (1), each tree $f_k$ is determined by its split structure "q" and node weights "w". The objective function for model optimization is Equation (2).

$$L(f_k) = \sum_{i=1}^{n} l(y_i, \hat{y}_i) + \sum_{k=1}^{K} \Omega(f_k)$$

(2)

In Equation (2), "l" denotes a twice-differentiable loss function, and $\Omega(f) = \gamma T + \frac{1}{2}\lambda \sum_{j=1}^{T} w_j^2$ represents the regularization term for each tree, with $\gamma$ and $\lambda$ controlling model complexity. During training, the model is updated iteratively in the (t)-th round according to Equation (3), and the objective function at the current round, Equation (4), is approximated by a second-order Taylor expansion at $f_t$, resulting in Equation (5).

$$\hat{y}_i^{(t)} = \hat{y}_i^{(t-1)} + f_t(x_i)$$

(3)

$$L^{(t)} = \sum_{i=1}^{n} \left( \hat{y}_i^{(t-1)} + f_t(x_i) \right) + \Omega(f_t)$$

(4)

$$L^{(t)} \approx \sum_{i=1}^{n} \left[ g_i f_t(x_i) + \frac{1}{2} h_i f_t(x_i)^2 \right] + \Omega f_t$$

(5)

In Equation (5), $(g_i)$ represents the first-order derivative of the loss function, and $(h_i)$ represents the second-order derivative. $g_i = \frac{\partial l\left(y_i, \hat{y}_i^{(t-1)}\right)}{\partial \hat{y}_i^{(t-1)}}$, $h_i = \frac{\partial^2 l\left(y_i, \hat{y}_i^{(t-1)}\right)}{\partial^2 \hat{y}_i^{(t-1)}}$. The algorithm relies on the $g_i$ and $h_i$ statistics to evaluate candidate split gains, selecting the optimal split through a greedy search. Leaf node numbers and weight regularization are controlled via $\gamma$ and $\lambda$, enhancing predictive performance while mitigating overfitting.

## ICE model

The Individual Conditional Expectation (ICE) model serves as a crucial interpretative tool for tree models. Compared with partial dependence plots (PDPs), which describe average effects, ICE plots visualize how the predicted outcome for each

individual sample responds to changes in a target built-environment variable, thereby providing a more detailed representation of local heterogeneity within the sample set. However, the spatial characteristics of ICE results are often difficult to interpret directly [50]. Therefore, this study defines $\Delta y^n$ as the variation in the predicted values along the ICE curve for the $n$-th sample to quantify the significance of threshold effects at the individual level. The calculation is expressed as follows:

$$\Delta y^n = max\{y(x_z, x_c^{(n)})\} - min\{y(x_z, x_c^{(n)})\} \tag{6}$$

In equation (6), $y(x_z, x_c^{(n)})$ denotes the predicted value for a given sample, $x_z$ represents the target explanatory variable, and $x_c^n$ refers to the observed values of the remaining variables for the $n$-th sample. The values of $x_z$ are varied across the full range of observed samples of the target explanatory variable. The indicator $\Delta y^n$ captures the significance of the threshold effect: a larger value of $\Delta y^n$ indicates a more pronounced threshold effect of the built-environment factor on residents' willingness to commute by the Loop Line Railway. Furthermore, the spatial distribution of $\Delta y^n$ reflects the spatial pattern of the strength of threshold effects associated with built environment factors.

On the basis of quantifying the significance of threshold effects, this study further defines "$m$" as the ratio of $\Delta y^n$ to $\Delta x_z^n$, where $\Delta x_z^n$ represents the variation in the sample value of the target explanatory variable corresponding to the maximum and minimum predicted values of sample "n". The calculation is expressed as follows:

$$m = \frac{\Delta y^n}{\Delta x_z^n} \tag{7}$$

In equation (7), the indicator "$m$" is derived from the ICE curves. A larger value of "$m$" indicates a stronger degree of non-linear influence of the built-environment factor on residents' willingness to commute by the Loop Line Railway. Specifically, the "$m$" captures the non-linear effect of an individual built-environment variable on commuting willingness. By visualizing the spatial distribution of "$m$", the spatial heterogeneity of such non-linear effects can be intuitively revealed. Furthermore, the spatial patterns of "$m$" provide a basis for examining the spatial non-stationarity in the dependence of commuting willingness on built-environment characteristics.

## Analysis of results

### Feature variable contribution

The SHAP model is employed in this study to analyze the significance of various built environment variables. As indicated in Fig 3a, population density, distance to the nearest bus stop, road network density, and distance to the city center emerge as the four most influential factors affecting commuting willingness. Fig 3b presents the overall SHAP plot for built environment variables, where the vertical axis represents variable importance ranking and the horizontal axis denotes the SHAP value of the independent variable. A broader distribution of SHAP values corresponds to greater influence. Each point in the figure represents an individual observation, with color denoting the magnitude of the feature value: red indicates higher values, while blue denotes lower values [51]. Fig 3b reveals that population density and road network density positively influence Loop Line Railway commuting willingness, whereas distance to nearest bus stop and distance from city center exert negative effects.

### Significance of the threshold effect and its spatial distribution

Four variables—population density, distance to the nearest bus stop, road network density, and distance from the city center—were identified as key influencing factors based on the XGBoost feature importance and SHAP analysis results. Therefore, this paper analyses the threshold effects of these four factors on commuting willingness for the Loop Line Railway. Fig 4 presents the ICE diagram for these four variables, where the thick blue solid line represents the average value across all samples, and the bar chart illustrates the frequency distribution of sample values. As shown

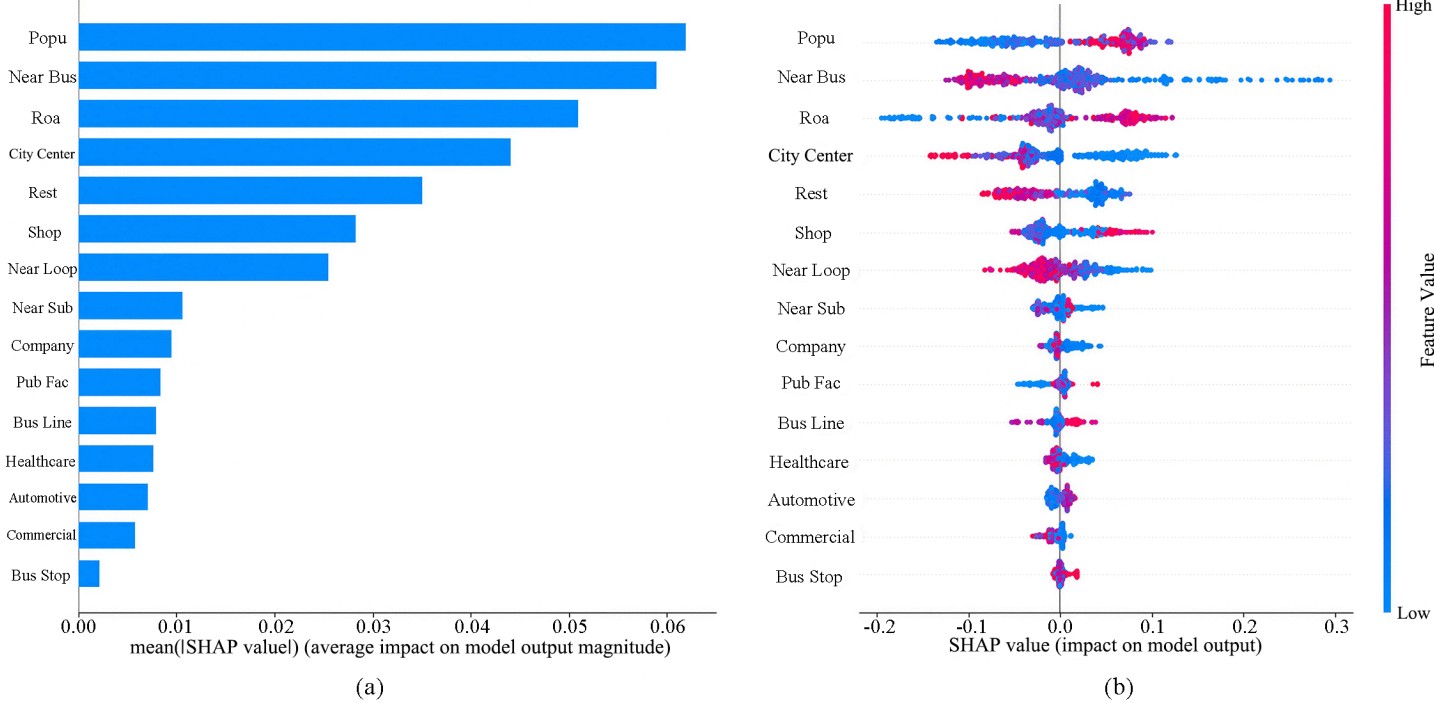

**Fig 3. Importance of Built Environment Factors and SHAP Values.** Note: a Variable Feature Importance; b SHAP values. This figure was produced by the author using python.

in Fig 4, the trend corresponding to the sample distribution range is consistent with the overall pattern. It should be noted that, according to the Theory of Planned Behavior (TPB) and the Technology Acceptance Model (TAM), objective environmental attributes influence behavioral intention primarily by shaping individuals' perceptions, attitudes, and perceived behavioral control [52]. In the context of suburban railway commuting, built environment conditions such as accessibility, connectivity, and spatial location may alter residents' perceived usefulness, perceived ease of use, and perceived convenience of the Loop Line Railway, which subsequently affect their commuting willingness. Therefore, the non-linear relationships identified in this section can also be understood as behavioral responses mediated through psychological mechanisms.

Research indicates that population density positively influences public transport usage frequency, yet this effect diminishes beyond a certain threshold, potentially prompting residents to switch to alternative transport modes [53]. As illustrated in Fig 4a, commuting willingness increases progressively with rising population density, with a noticeable threshold emerging around 20,000 inhabitants per square kilometer. This arises because commuting willingness progressively increases with population density, thereby strengthening the appeal of commuting via the Loop Line Railway system [54]. Conversely, at stations with lower population density, the smaller population base and insufficient commuting willingness hinder the system's commuting functionality [55]. However, indiscriminately increasing population density near stations diminishes the system's comfort and attractiveness, ultimately reducing commuting willingness [56]. From a behavioral perspective, population density also influences commuting willingness through psychological pathways. Higher density is often associated with better service frequency, stronger social visibility of public transport use, and more pronounced collective travel norms, which may enhance subjective norms and perceived usefulness of the Loop Line Railway [57]. In contrast, excessive density may reduce perceived comfort and negatively affect attitudes toward rail commuting [58].

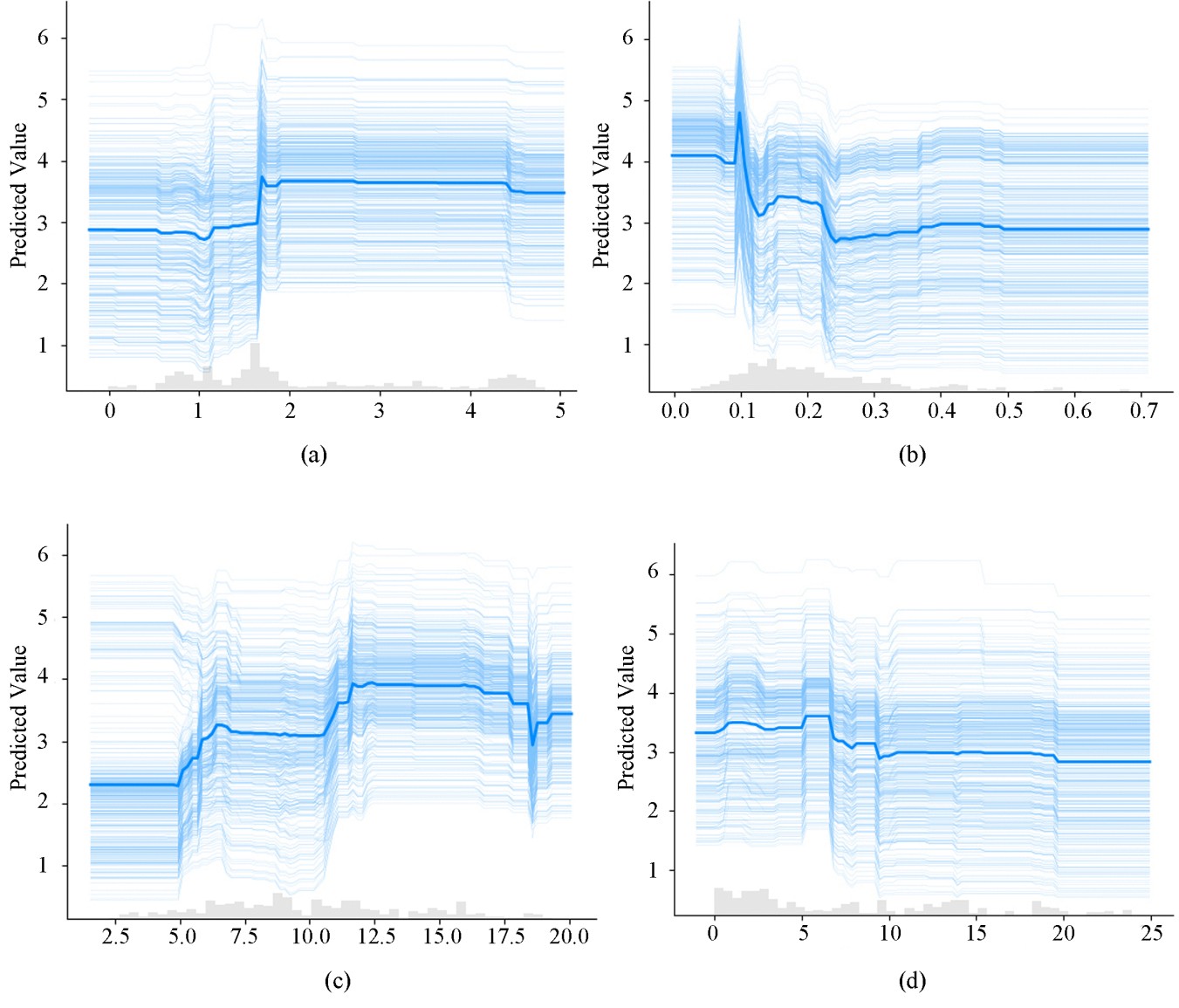

**Fig 4. ICE Model of Commuting Willingness for Loop Line Railway Based on Key Built Environment Factors.** Note: a Population Density(10,000 people/km²); b Distance to Nearest Bus Stop(km); c Road Network Density(km/km²); d Distance to City Center(km). This figure was produced by the author using python.

Therefore, the observed threshold effect reflects not only spatial constraints but also changes in psychological perceptions and attitudes, consistent with TPB and TAM mechanisms.

Previous studies have shown that the distance to the nearest bus stop influences residents' travel decisions by affecting transfer convenience [59]. This study further finds that the distance to the nearest bus stop indirectly affects residents' willingness to commute by the Loop Line Railway by shaping first- and last-mile access conditions to railway stations. As shown in Fig 4b, residents' willingness to use the Loop Line Railway for commuting generally declines as the distance from their residence to the nearest bus stop increases, and a clear threshold effect is observed around 0.2 km. It should be noted that in commuter rail or suburban railway systems, travelers often rely on multiple access modes for first- and

last-mile connections (e.g., buses, cycling, and other non-motorized modes), rather than walking alone. This indicates a higher sensitivity to public transport connection conditions than in urban metro systems. For example, in a study of suburban rail commuting in Dhaka by Rahman [60], it was found that multimodal combinations are widely used for first- and last-mile access. Therefore, the distance to the nearest bus stop effectively serves as a proxy for the adequacy of public transport feeder connections around Loop Line Railway stations. Given the relatively large station spacing and wide service coverage of the Loop Line Railway, residents tend to rely on buses and other public transport modes to complete first- and last-mile trips, with the bus system playing an important feeder and distribution role. Consequently, proximity to a bus stop correlates with reduced time costs and uncertainty associated with accessing the Loop Line Railway during commuting. This facilitates the perception and realization of the railway's advantage as a "rapid commuting" mode, thereby enhancing its attractiveness as a commuting option. From the perspective of behavioral theory, improved feeder accessibility can enhance perceived ease of use and perceived behavioral control by reducing transfer difficulty and uncertainty [61]. When residents perceive multimodal connections to be convenient and reliable, their confidence in using the Loop Line Railway increases, thereby strengthening commuting intention. Conversely, longer distances to feeder transit may reduce perceived accessibility and increase psychological resistance to rail commuting. This mechanism further demonstrates that built environment factors influence commuting willingness through perception-based pathways, consistent with TPB constructs.

The predicted relationship between road network density and commuting willingness is illustrated in Fig 4c. A non-linear relationship between road network density and commuting willingness is indicated by the results, with willingness gradually increasing as road network density rises. A threshold effect is observed around a network density of approximately 10 km/km². This phenomenon may be attributed to the fact that increased network density enhances access options to stations, improves station accessibility, and facilitates transfers, thereby boosting commuting willingness [62]. However, excessive increases in network density may elevate commuting costs [63]. Moreover, greater use of alternative transport modes may be encouraged by denser networks, potentially diminishing the appeal of the Loop Line Railway. Consequently, areas with moderate network density should be prioritized in site selection to optimize utilization rates. Behaviorally, increased road connectivity may enhance perceived convenience and perceived behavioral control associated with reaching stations [64]. However, overly developed road systems may strengthen the perceived competitiveness of private cars or ride-hailing services, thereby weakening the perceived usefulness of rail commuting. This suggests that the observed non-linear relationship reflects changes in comparative modal perception rather than purely infrastructural effects.

Residents' commuting willingness is influenced by the distance from the city center. As shown in Fig 4d, willingness to commute via the Loop Line Railway diminishes as the distance from the city center increases. This trend may be attributed to the well-developed transport infrastructure and convenient connections near the city center. Conversely, reduced transport accessibility, longer transfer and commuting times, and increased commuting costs are experienced in areas farther from the urban core.Residents in such areas become more reliant on private vehicles or alternative transport modes, leading to diminished willingness to commute [59]. From a psychological perspective, spatial distance from the city center influences perceived usefulness and perceived effort associated with commuting. Residents located farther from employment centers may perceive higher travel burdens and uncertainty, which negatively affect attitudes toward rail commuting. However, in certain contexts, the rapid travel advantage of suburban rail may partially offset this spatial disadvantage, thereby enhancing perceived usefulness. These findings indicate that spatial location influences commuting willingness through cognitive evaluations of time savings and travel convenience.

As illustrated in Fig 4, although the four built environment categories exert differing degrees of influence on commuting willingness, all observations exhibit non-linear characteristics at the threshold, suggesting a consistent threshold effect across the entire dataset. When threshold effects are present, the magnitude of the change in predicted values (Δy) reflects the significance of the threshold effect for a specific observation. A larger (Δy) indicates a more pronounced

threshold effect for the corresponding observation. The spatial distribution of (Δy) reveals the spatial heterogeneity in threshold-effect significance across various locations. Fig 5 presents the spatial distribution of threshold-effect intensity for the key built environment factors that influence commuting willingness via the Loop Line Railway. The results reveal substantial spatial heterogeneity in threshold-effect strength across all examined variables.

Fig 5a indicates that (Δy) values for population density exhibit a decreasing spatial gradient from the city center to suburban and peripheral regions. High (Δy) values are concentrated near Guiyang Station and Guiyang North Station; conversely, high-value clusters are less prevalent in suburban locations (e.g., Baiyun North and Jinyang South stations) and peripheral areas (e.g., Huaxi South and Gui'an stations), where the distribution of (Δy) appears more localized. This pattern is driven by the high population density and functional agglomeration of central urban areas, where changes in

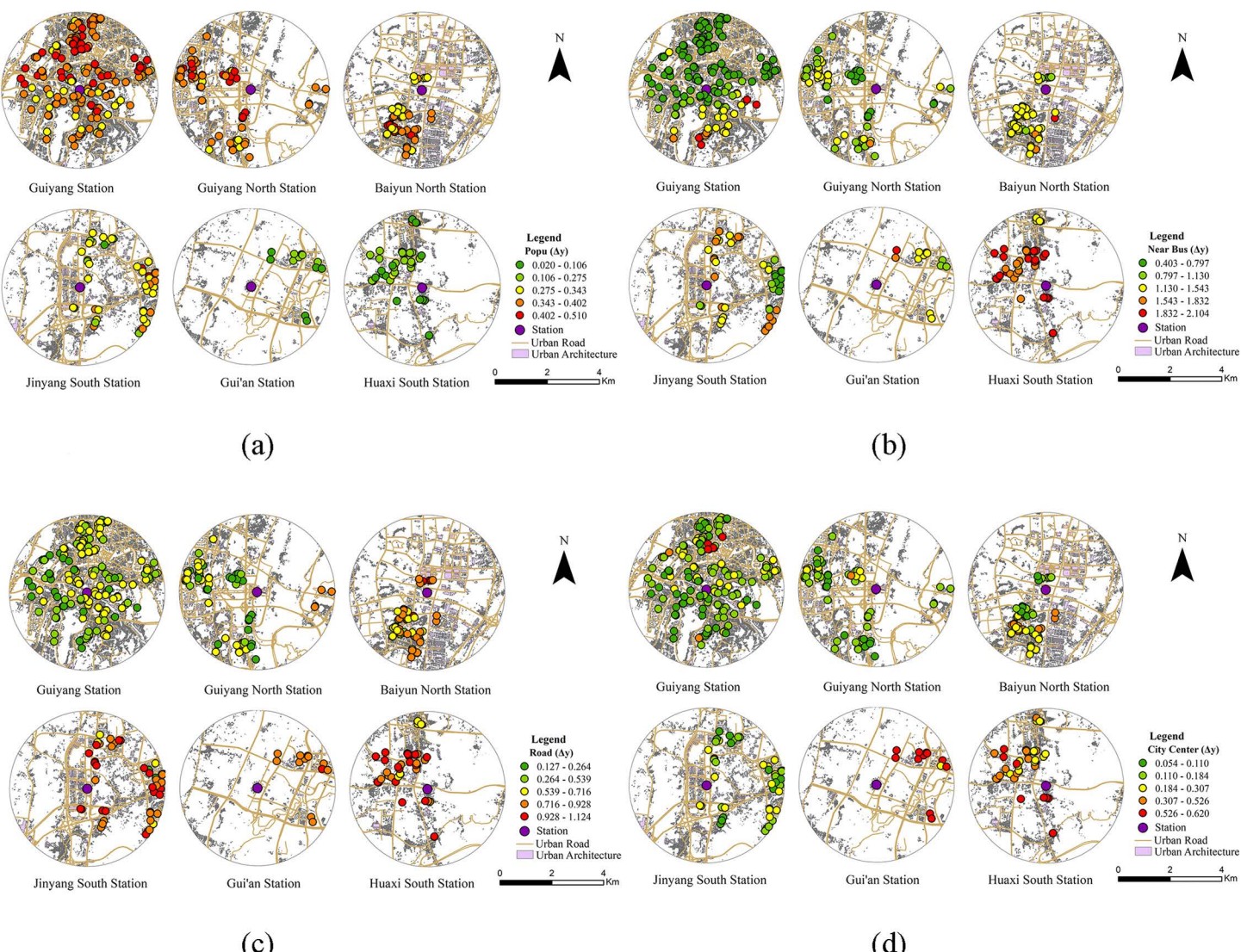

**Fig 5. Spatial Distribution of Significant Threshold Effects for Key Built Environment Factors.** Note: a Population density (10,000 persons/km²); b Distance to the nearest bus stop (km); c Road network density (km/km²); d Distance to city center (km). This figure was produced by the author using ArcGIS.

population scale trigger concurrent variations in commuting willingness and travel pressure, thereby accentuating threshold effects. In contrast, lower population density and dispersed residential development in peripheral areas result in a smaller marginal influence of population variations on commuting decisions, leading to less pronounced threshold effects.

Fig 5b depicts the spatial distribution of (Δy) values for distance to the nearest bus stop. High (Δy) values are primarily concentrated in suburban areas such as Huaxi South Station, with scattered occurrences in peri-urban locations, including Baiyun North and Jinyang South stations. This spatial pattern suggests that residents in urban fringe areas with relatively weak public transport networks exhibit greater sensitivity to changes in bus-rail transfer conditions. In areas with sparse bus stop distribution or greater distances from residential zones, the higher transfer costs and uncertainty associated with the Loop Line Railway weaken its attractiveness as a commuting mode. Conversely, improvements in feeder transit conditions (e.g., enhanced bus coverage or seamless bus-rail integration) are associated with more pronounced threshold effects, underscoring the critical role of bus accessibility in first- and last-mile travel within peripheral areas.

Fig 5c depicts variations in road network density (Δy) values. The map reveals that regions with pronounced threshold effects (Δy) are predominantly concentrated around Baiyun North, Jinyang South, Gui'an, and Huaxi South stations. This pattern indicates that where road infrastructure is insufficient, variations in road network density trigger more substantial changes in commuting willingness. In these regions, enhanced road connectivity (e.g., reduced detour distances or increased route options for accessing stations) are likely to improve overall accessibility and travel efficiency, thereby increasing the perceived feasibility of commuting via the Loop Line Railway. In contrast, the well-developed and highly connected road networks of central urban areas mean that marginal changes in density exert limited influence on commuting decisions.

Fig 5d shows that the threshold effects of "distance to the city center" are more pronounced at stations located farther from the urban core (e.g., Huaxi South and Gui'an). This result suggests that in peripheral urban areas, spatial location differences between residential areas and the city center play a stronger role in shaping residents' commuting decisions. It should be noted that variations in the (Δy) values associated with distance to the city center primarily reflect the heightened sensitivity of residents in peripheral areas to commuting distance and related travel costs. When residents are situated far from the urban core, factors that mitigate this "spatial distance disadvantage"—such as the rapid accessibility provided by the Loop Line Railway—exhibit an amplified influence on commuting willingness.

## Spatial characteristics of non-linear effects

Residents at stations characterized by built environment threshold effects demonstrate non-linear variations in commuting willingness. This non-linear dependence of commuting willingness on the built environment exhibits significant spatial effects. These characteristics are reflected in the spatial distribution of m-values, as illustrated in Figs 6–9. The spatial distribution of (m) aligns with the observed (Δy) distribution, further confirming significant spatial heterogeneity. This demonstrates that the non-linear dependence of commuting willingness on the built environment exhibits spatial non-stationarity.

Fig 6 illustrates the spatial distribution of the non-linear impact exerted by population density on commuting willingness. Comparing the six stations reveals that for central stations such as Guiyang Station, Guiyang North Station, and the southern areas of Baiyun North Station, where population density is high and commuting willingness is significant, the non-linear effect is more pronounced. Conversely, at suburban stations like Gui'an Station and Huaxi South Station, where population density is low and commuting willingness is weak, the non-linear effect is not significant. Previous research indicates that population density exerts a non-linear threshold effect on individual mode choice [38]. This study further reveals that this non-linear impact exhibits spatial heterogeneity, characterized by a decreasing gradient from the urban core to the periphery. This suggests that, relative to lower-density suburban stations, central urban areas generate greater potential willingness through the clustering of population and employment. Consequently, the appeal of the Loop Line Railway as a commuting option is amplified for these residents. Therefore, station-siting strategies should be tailored to

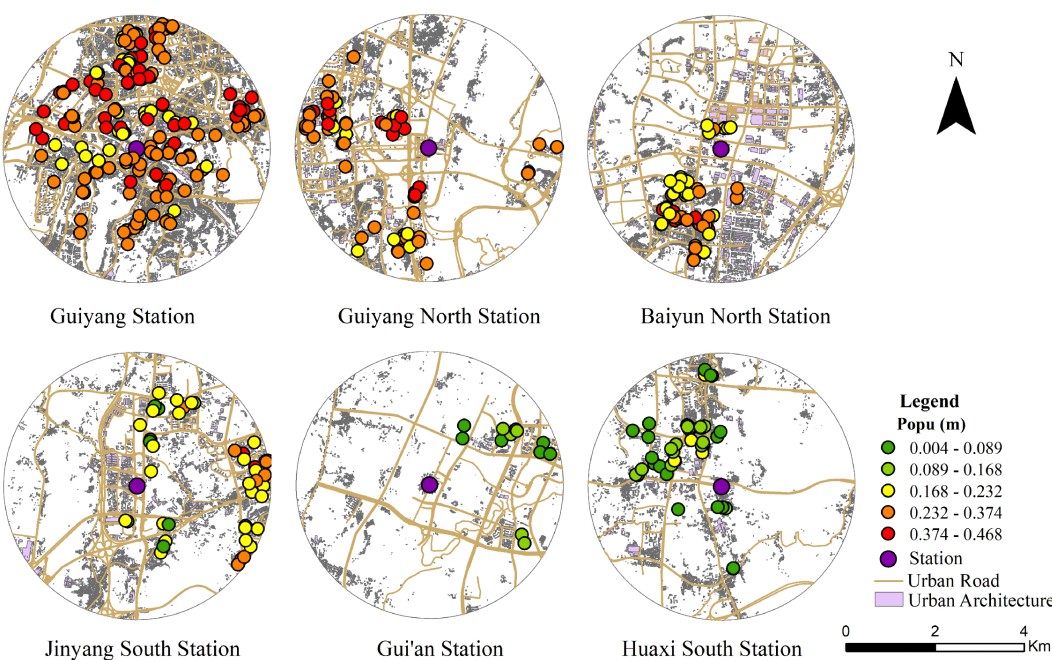

**Fig 6. Spatial distribution of the non-linear effect of population density on commuting willingness for the Loop Line Railway.** This figure was produced by the author using ArcGIS.

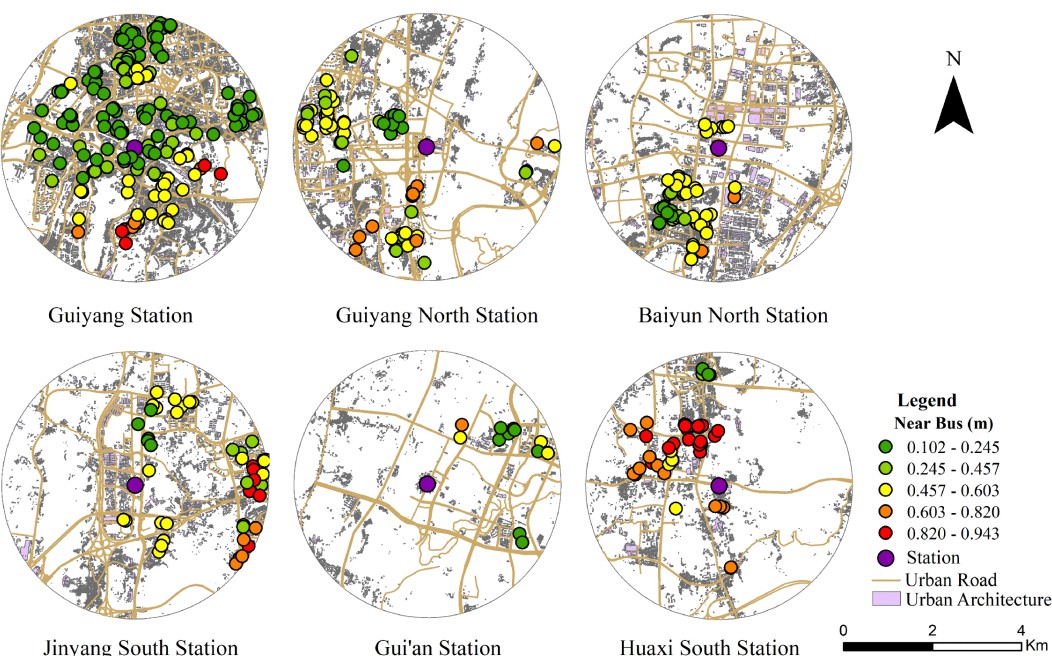

**Fig 7. Spatial distribution of the non-linear effect of nearest bus stop distance on commuting willingness for the Loop Line Railway.** This figure was produced by the author using ArcGIS.

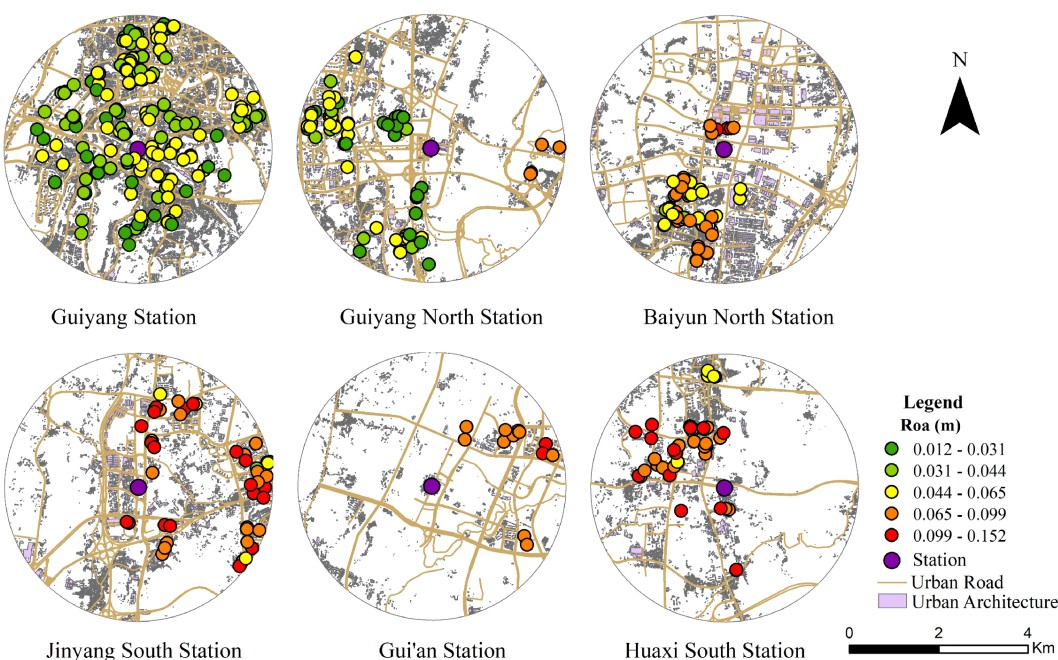

**Fig 8. Spatial distribution of the non-linear effect of road network density on commuting willingness for the Loop Line Railway.** This figure was produced by the author using ArcGIS.

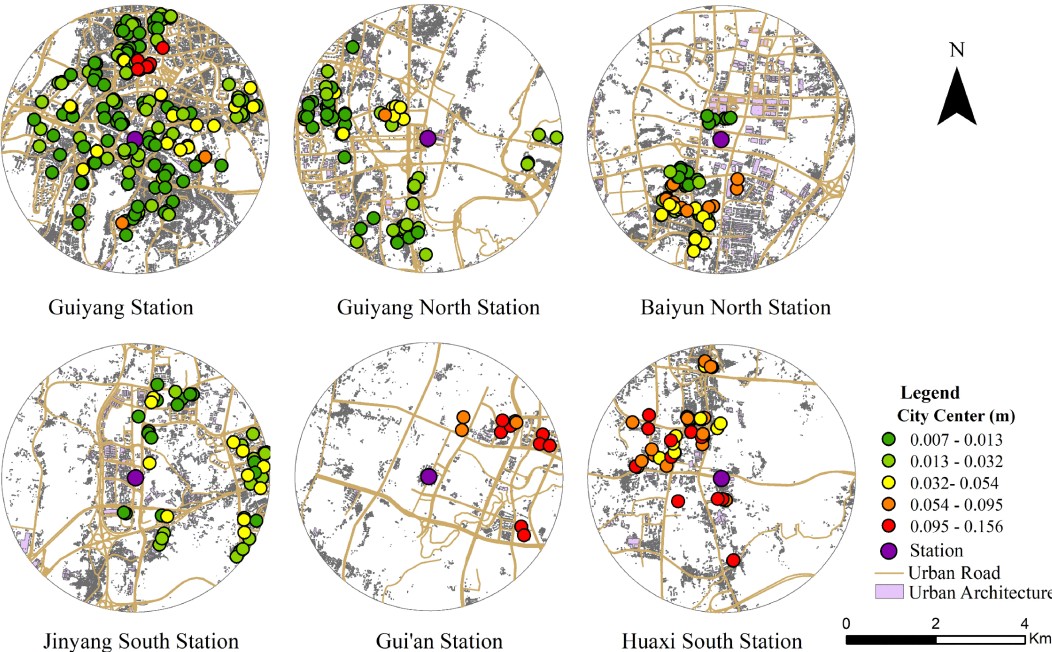

**Fig 9. Spatial distribution of the non-linear effect of distance from city center on commuting willingness for the Loop Line Railway.** This figure was produced by the author using ArcGIS.

regional characteristics, prioritizing high-density areas—such as jobs-housing balance zones—to enhance railway ridership [65].

Previous studies investigating the influence of public transport on travel mode choice have primarily focused on indicators such as bus stop density, bus route coverage, and the overall scale of bus service provision to explain how bus systems shape residents' travel behavior [66]. By contrast, this study adopts a more granular indicator—the distance to the nearest bus stop—and reveals a significant non-linear effect on commuting willingness, characterized by pronounced spatial heterogeneity. This effect is particularly prominent at suburban and peripheral stations. As illustrated in Fig 7, the distance to the nearest bus stop exerts a more significant non-linear effect at suburban hubs (e.g., Jinyang South Station) and outlying stations (e.g., Huaxi South Station). This suggests that in peripheral urban areas with sparse bus networks and limited coverage, bus accessibility becomes a critical constraint governing residents' decisions to commute via the Loop Line Railway. Although the survey data preclude a direct quantification of the proportion of intermodal commuters, the high sensitivity of commuting willingness to bus proximity in suburban and peripheral areas suggests that buses primarily function as feeder services rather than independent travel modes. This reflects the accessibility conditions under which residents obtain public transport connections. The findings further indicate that, in peripheral areas with limited bus resources, shortening the distance for residents to reach the nearest bus stop and reducing the time cost and uncertainty of the first- and last-mile can effectively enhance the commuting appeal of the Loop Line Railway.

A higher road network density signifies well-developed road infrastructure, enhanced station accessibility, and improved transfer efficiency, which are crucial for promoting the utilisation of the Loop Line Railway [65]. Guiyang is a typical mountainous city where topographical constraints yield a high road circuity coefficient. Consequently, the effective walking radius is often significantly constrained by elevation differences and the circuitous nature of the road network [67]. Accordingly, road network density significantly influences residents' propensity to commute via the Loop Line Railway. As illustrated in Fig 8, stations where road network density exerts a significant non-linear impact are primarily concentrated near Baiyun North, Jinyang South, Gui'an, and Huaxi South stations. These regions are characterized by sparse road networks, poor accessibility, and inconvenient transfers, which often deter residents from choosing the Loop Line Railway [39]. Therefore, appropriately enhancing the connectivity and accessibility of the road network surrounding the station, increasing the number of alternative routes and interchange options available, will reduce barriers to residents commuting via the Loop Line Railway. This will consequently enhance commuting willingness for the Loop Line system. Conversely, in areas such as Guiyang Station and Guiyang North Station, where road network density is high and the road network is well-developed, further increasing road network density has a weak or even inhibitory effect on promoting the willingness to commute using the Loop Line Railway.

Consistent with previous studies, residents' propensity to commute varies with distance from the urban core, a relationship influenced by station connectivity, supporting facilities, and service frequency [37]. This paper further identifies distinct spatial disparities regarding the potential to enhance commuting willingness across central urban and suburban zones. As illustrated in Fig 9, significant non-linear effects of distance on commuting willingness are primarily concentrated around Gui'an and Huaxi South stations. These areas are located in the urban periphery and are relatively far from the city center. In such contexts, the Loop Line Railway outperforms alternative modes in travel time, operating velocity, and cross-regional accessibility. Consequently, residents in remote locations can complete commutes within acceptable timeframes, thereby reinforcing their inclination to utilize the Loop Line Railway for daily travel. Conversely, regions such as Guiyang Station and Guiyang North Station, being closer to the city centre, exhibit limited potential for further increasing commuting willingness via the Loop Line Railway network through reduced travel distances, resulting in a less pronounced nonlinear effect. Consequently, when formulating Loop Line Railway policies, priority should be given to peripheral areas, providing a rapid and efficient commuting option for regions distant from the city centre.

 

## Discussion and policy implications

Drawing on the spatial analysis of threshold effects and non-linear impacts of built environment factors, this study proposes the following recommendations to address the low ridership and underutilisation of the Guiyang Loop Line Railway, and to provide practical insights for station planning and the development of suburban railways in other cities.

(1) The results indicate that the commuting function of the Loop Line Railway depends heavily on the presence of stable and sufficiently concentrated commuting willingness. In areas with high population density and strong job-housing concentration, the Loop Line Railway is more suitable as a frequent and reliable commuting mode, and its commuter-oriented function should be prioritised. In contrast, in peripheral areas characterised by lower population density and dispersed residential patterns, changes in population scale alone are unlikely to meaningfully enhance the commuting attractiveness of the Loop Line Railway. In such contexts, improving feeder connections and reducing overall commuting time represent more effective strategies for strengthening its relative advantages.

(2) For suburban and peripheral stations, priority should be given to improving the accessibility of bus feeder services rather than expanding the overall scale of bus supply. Rather than increasing the number of bus routes, more effective measures include optimising the spatial distribution of bus stops, shortening walking distances to the nearest bus stop, and providing dedicated feeder bus services to railway stations. These measures can reduce first- and last-mile transfer costs and, in turn, enhance the commuting attractiveness of the Loop Line Railway.

(3) The model results show that the non-linear effects of road network density on commuting willingness are mainly concentrated in suburban and peripheral areas. Accordingly, road network improvements should focus on areas where transport infrastructure is relatively underdeveloped. Measures such as improving local street networks, reducing detour distances, and enhancing road connectivity can increase station accessibility and facilitate residents' access to railway stations, thereby encouraging the use of the Loop Line Railway for commuting. In contrast, in central urban areas where road networks are already highly developed, further increases in road density may yield limited benefits. Instead, measures such as dedicated access lanes for railway transfers and enhanced traffic flow management can be adopted to reduce congestion and improve transfer efficiency.

(4) The Loop Line Railway plays an important role in mitigating the disadvantages associated with long commuting distances in peripheral areas. For residents living far from the urban core, commuting distance and time costs constitute major constraints on railway use. In these areas, the cross-regional rapid accessibility provided by the Loop Line Railway is more likely to be strongly perceived and valued. Accordingly, in the process of optimising urban spatial structure, the Loop Line Railway can serve as an important instrument for improving commuting conditions in peripheral areas and promoting shifts in commuting modes. However, given the relatively low level of development in these areas, particular attention should be paid to strengthening feeder transport facilities and reducing overall commuting time. From a behavioral perspective, optimizing built environment conditions around stations not only improves objective accessibility but also enhances perceived usefulness and perceived behavioral control, thereby strengthening residents' willingness to use suburban railways for commuting.

## Conclusions and limitations

### Conclusions

Based on survey data on residents' willingness to commute via the Guiyang Loop Line Railway, this study systematically examines the non-linear effects of built environment factors on commuter willingness and their spatial heterogeneity. The main conclusions are as follows:

 

(1) Population density, distance to the nearest bus stop, road network density, and distance from the city center represent the four built environment factors with the most substantial impact on commuting willingness. Specifically, population density and road network density exert a positive influence, whereas distance to the nearest bus stop and distance from the city center show an inverse correlation with commuting willingness for the Loop Line Railway.

(2) The non-linear dependence of commuting willingness on the built environment exhibits strong spatial non-stationarity along the Loop Line Railway. The significance of threshold effects and the intensity of non-linear impacts associated with different built environment factors vary substantially across space, forming a clear gradient from the urban core to suburban and peripheral areas. Compared with the central urban area, residents living around suburban and peripheral stations show greater sensitivity to changes in feeder bus conditions, road network connectivity, and commuting distance, which results in more pronounced threshold effects and stronger non-linear responses.

(3) The mechanisms through which built environment factors influence commuting willingness differ across spatial contexts, with peripheral areas showing greater potential for improving commuting willingness. Specifically, the threshold effects and non-linear impacts of population density are primarily concentrated around central urban stations, indicating that in areas with highly concentrated commuting willingness, changes in population scale have stronger marginal effects on willingness to commute via the Loop Line Railway. In contrast, the non-linear effects of distance to the nearest bus stop and road network density are more pronounced around suburban and peripheral stations, indicating that in areas with relatively underdeveloped public transport and road infrastructure, first- and last-mile connection conditions and station accessibility represent key constraints on using the Loop Line Railway for commuting. In addition, the non-linear effect of distance to the city center is most pronounced at stations located farther from the urban core, suggesting that the Loop Line Railway has greater potential to mitigate the spatial disadvantage associated with long commuting distances in peripheral areas.

(4) Furthermore, the findings of this study also provide empirical support for the behavioral mechanisms proposed by the Theory of Planned Behavior (TPB) and the Technology Acceptance Model (TAM). The results indicate that built environment characteristics do not influence commuting willingness solely through physical accessibility constraints, but also indirectly by shaping residents' perceptions of convenience, usefulness, and travel effort. Improvements in population density, feeder accessibility, and road connectivity may enhance perceived usefulness, perceived ease of use, and perceived behavioral control associated with the Loop Line Railway, thereby strengthening commuting intention. Conversely, unfavorable spatial conditions may reduce perceived advantages and increase psychological resistance to rail commuting. These findings highlight the importance of integrating objective spatial planning with perception-oriented service improvements to promote suburban railway utilization.

## Limitations and future research directions

Although this study provides a systematic analysis of the non-linear effects and spatial heterogeneity of built environment factors on commuting willingness, several limitations remain that warrant further investigation.

First, this research is limited to the Guiyang Loop Line Railway as a single case study. As a typical mountainous city, Guiyang possesses distinctive topographical features, urban spatial structures, and a unique rail transit development stage. Therefore, caution is required when generalizing the findings to cities with divergent geographical conditions or development trajectories. Future studies should incorporate comparative analyses across multiple cities or railway lines to validate the robustness and generalizability of the conclusions.

Second, the analysis relies on cross-sectional survey data, which captures residents' stated commuting willingness at a single temporal point. Consequently, the study cannot fully capture dynamic shifts in commuting willingness resulting from built environment improvements or operational adjustments. Future research could benefit from integrating longitudinal

survey data with objective travel data (e.g., smart card or mobile phone signaling data) to better examine the long-term behavioral impacts of built environment interventions.

Finally, data constraints limited the built environment measurement to indicators such as population density, road network density, and bus stop proximity. Other potentially relevant factors, such as pedestrian environment quality and public transport reliability, were not explicitly considered. Future studies should incorporate finer-grained spatial and environmental data to elucidate the mechanisms through which the built environment influences suburban railway commuting behavior.

## Author contributions

**Conceptualization:** Rufeng Chen, Hang Zhao.

**Data curation:** Rufeng Chen.

**Formal analysis:** Rufeng Chen.

**Funding acquisition:** Rufeng Chen, Hang Zhao.

**Investigation:** Rufeng Chen.

**Methodology:** Rufeng Chen.

**Project administration:** Rufeng Chen, Hang Zhao.

**Resources:** Rufeng Chen.

**Software:** Rufeng Chen.

**Supervision:** Rufeng Chen, Hang Zhao.

**Validation:** Rufeng Chen.

**Visualization:** Rufeng Chen.

**Writing – original draft:** Rufeng Chen.

**Writing – review & editing:** Rufeng Chen, Hang Zhao.

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
