## [Decision Letter · Decision Letter 0]

28 Dec 2025

Dear Dr. Zhao,

Thank you for submitting your manuscript to PLOS ONE. After careful consideration, we feel that it has merit but does not fully meet PLOS ONE’s publication criteria as it currently stands. Therefore, we invite you to submit a revised version of the manuscript that addresses the points raised during the review process.

We look forward to receiving your revised manuscript.

Kind regards,

Qing-Chang Lu

Academic Editor

PLOS One

Journal Requirements:

3. Please note that PLOS One has specific guidelines on code sharing for submissions in which author-generated code underpins the findings in the manuscript. In these cases, we expect all author-generated code to be made available without restrictions upon publication of the work. Please review our guidelines at https://journals.plos.org/plosone/s/materials-and-software-sharing#loc-sharing-code and ensure that your code is shared in a way that follows best practice and facilitates reproducibility and reuse.

4. In your Methods section, please provide additional information regarding the permits you obtained for the work. Please ensure you have included the full name of the authority that approved the field site access and, if no permits were required, a brief statement explaining why.

5. Please provide a complete Data Availability Statement in the submission form, ensuring you include all necessary access information or a reason for why you are unable to make your data freely accessible. If your research concerns only data provided within your submission, please write "All data are in the manuscript and/or supporting information files" as your Data Availability Statement.

6. We note that you have indicated that there are restrictions to data sharing for this study. PLOS only allows data to be available upon request if there are legal or ethical restrictions on sharing data publicly. For more information on unacceptable data access restrictions, please see http://journals.plos.org/plosone/s/data-availability#loc-unacceptable-data-access-restrictions.

7. Thank you for stating the following in the Acknowledgments Section of your manuscript:

The research is supported by the National Natural Science Foundation of China [Grant No. 71864008].The funder website is https://www.nsfc.gov.cn/. The funder  did not participate in the research design, data collection and analysis, publication decisions, or manuscript writing.

The research is supported by the National Natural Science Foundation of China [Grant No. 71864008].

8. Please update your submission to use the PLOS LaTeX template. The template and more information on our requirements for LaTeX submissions can be found at http://journals.plos.org/plosone/s/latex.

9. Please amend the manuscript submission data (via Edit Submission) to include author Xiaohuan Wang, Yahao Guo, Xiwei Xiong.

10. We note that Figure(s) 1, 2, 4, 5, 6, 7, and 8, in your submission contain [map/satellite] images which may be copyrighted. All PLOS content is published under the Creative Commons Attribution License (CC BY 4.0), which means that the manuscript, images, and Supporting Information files will be freely available online, and any third party is permitted to access, download, copy, distribute, and use these materials in any way, even commercially, with proper attribution. For these reasons, we cannot publish previously copyrighted maps or satellite images created using proprietary data, such as Google software (Google Maps, Street View, and Earth). For more information, see our copyright guidelines: http://journals.plos.org/plosone/s/licenses-and-copyright.

a. You may seek permission from the original copyright holder of Figure(s) 1, 2, 4, 5, 6, 7, and 8, to publish the content specifically under the CC BY 4.0 license.

11. Please ensure that you refer to Figure 1 in your text as, if accepted, production will need this reference to link the reader to the figure.

Reviewers' comments:

Reviewer's Responses to Questions

**Comments to the Author**

1. Is the manuscript technically sound, and do the data support the conclusions?

Reviewer #1: Partly

Reviewer #2: Yes

2. Has the statistical analysis been performed appropriately and rigorously?

Reviewer #1: Yes

Reviewer #2: No

3. Have the authors made all data underlying the findings in their manuscript fully available?

Reviewer #1: No

Reviewer #2: No

4. Is the manuscript presented in an intelligible fashion and written in standard English?

Reviewer #1: No

Reviewer #2: Yes

Reviewer #1: In the introduction, the authors stated that residents in Guiyang face limited transport options and poor road networks. Statistics and references are needed to support such claims.

What is Loop Line railway, and how it differs from other railways? This requires further explanation.

The underutilisation of the Loop Line railway is a very specific case. How it relates to the broader context and literature? The authors need to better position the current study in the literature.

The authors mentioned previous research on railway usage quite late into the introduction. The structure here can be considerably improved by reorganizing the logic, from broader picture to the specific case of Guiyang. It is also suggested to justify the case study as a valuable addition to the literature.

How are the sample and survey stations determined? This requires explanation and justification. This should bring forward in the methodological section.

It is unclear how the subjective statements are selected. It seems that some behavioural and social psychological theories might be considered. But a discussion of the relevant theories and the underlying framework are completely missing.

It is unclear how the proximity to bus stops may influence the usage of metro. The bus may serve as a feeder line to the railway. But it is unknown whether it is the case, and what is the proportion of intermodal transit users in Guiyang. The argued link between proximity to bus stops and walkability is weak by comparison.

It is unclear what Δy values refer to.

Results indicate that the effect of distance to city center is more pronounced in more remote areas, and suggests “improvements in distance conditions exert a substantial influence on residents' willingness to commute via the Loop Line Railway”. What improvements the authors refer to? This is very vague and needs specification. The same goes to the other three key variables.

It is unclear what m-values refer to.

There are some incomplete sentences, e.g., “In areas with lower bus stop density, establish industry development-oriented stations. Prioritise shuttle bus services or reduce transfer costs by increasing direct bus routes. Explore flexible transport options such as shared mobility to compensate for service gaps.” I won’t list all the examples, but this requires a careful proofread and revision.

In Spatial Analysis of Threshold Effects, the results are largely mixed with policy implications. As such the section is unfocused and difficult to follow. Many of the policy implications are also broad-brush and overly comprehensive. Some of them are not necessarily supported by the analysis here. The authors should (1) discuss the key findings and (2) derive policy implications in a separate section. It should be noted that the implications should be firmly based on the empirical analysis. Normally 3-5 key implications would suffice. At the moment, the added values and implications from the study are difficult to discern and evaluate.

Figures are of very low resolutions.

Reviewer #2: The data are generally reliable, and the model estimation is reasonable. However, the manuscript still requires substantial improvements in terms of detailed descriptions of the data and methodological procedures. The language should also be carefully revised to improve clarity and meet to standard academic languages.

.

Reviewer #1: No

Reviewer #2: No

---

## [Author Response · Author response to Decision Letter 1]

2 Feb 2026

Point-to-Point Responses to Reviewers and Editor

The authors are very thankful to the anonymous referees for the detail analysis and comments on the work. A number of interesting issues, questions and useful advises have been raised through the comments. In the past days, we have considered very carefully and seriously the questions and suggestions enclosed in the review and made modifications, adjustments, elaborations, etc. Accordingly, to make this paper more convincing, explicit, complete and acceptable. In the following the authors respond to each of their concerns and advises in an itemized fashion.

Responses to comments from Reviewer # 1

Reviewers comment 1

In the introduction, the authors stated that residents in Guiyang face limited transport options and poor road networks. Statistics and references are needed to support such claims.

Response to comment 1

We appreciate the reviewer’s constructive suggestions regarding the description of Guiyang’s current transportation status in the Introduction. To strengthen the empirical evidence and persuasiveness of our argument, we have incorporated reports from authoritative media outlets concerning Guiyang’s challenging road network conditions in the revised manuscript. The specific details are provided as follows (see Lines 60–70 in the revised manuscript):

It is indicated in the Guiyang “14th Five-Year” Special Transportation Construction Plan that the road network structure in the central urban area remains incomplete, characterized by a low proportion of secondary and branch roads, poor network circulation, and a severe shortage of overall road network density. Consequently, further improvement is required for the traffic carrying capacity [15]. By October 2024, an increase of approximately 20,000 motor vehicles per month was recorded in Guiyang, while the urban road network density stood at only 6.7 km/km², far below the national standard of 8 km/km². This imbalance between vehicle growth and road supply has led to a pronounced disparity between vehicle proliferation and road infrastructure availability, further constraining the efficiency of cross-regional commuting[16],

Reviewers comment 2

What is Loop Line railway, and how it differs from other railways? This requires further explanation.

Response to comment 2

We sincerely thank the reviewer for this insightful comment. We agree that the initial conceptual definition of the 'Loop Line Railway' and its distinction from other urban rail transit modes were not sufficiently articulated. In response, we have expanded the Introduction to include a more rigorous definition, functional positioning, and a comparative analysis of the Loop Line Railway versus other railway types. These additions aim to enhance conceptual clarity and the overall readability of the manuscript. The revised text is as follows (see Lines 2–17 in the revised manuscript)：

Suburban railways, often termed commuter rail or suburban rail, are categorized as a form of passenger rail system positioned between urban rail transit (e.g., metro systems) and intercity railways; they are characterized by commuter-oriented services, relatively high operating speeds, and large transport capacity. As a long-distance commuting mode, a crucial role is played by suburban railways in connecting central urban areas with peripheral suburbs and in supporting the development of a “one-hour commuting circle” [1]. In comparison to conventional urban rail transit, a wider spatial area is typically served by suburban railways, which feature larger station spacing, higher operating speeds, and lower service frequencies. They are primarily designed to accommodate medium-to long-distance commuting between central cities and surrounding suburban areas or satellite towns. Their functional orientation is focused on establishing efficient commuting corridors within metropolitan areas[2-3]. In this context, a significant contribution is made by suburban railways toward relieving functional pressure in central urban areas, shaping rational commuting structures, facilitating convenient daily mobility, and improving accessibility in peripheral urban regions [4].

Reviewers comment 3

The underutilisation of the Loop Line railway is a very specific case. How it relates to the broader context and literature? The authors need to better position the current study in the literature.

Response to comment 3

We sincerely thank the reviewer for this critical insight. We agree that the underutilization of the Guiyang Loop Line Railway should not be viewed as an isolated case but rather needs to be discussed within the broader context of regional and suburban rail research. Following this suggestion, we have systematically expanded the literature review in the Introduction to emphasize the universal significance of this study. Specifically, the revised manuscript now highlights that the underperformance of commuting functions in regional railways is a common challenge faced by multiple cities in China, rather than an issue unique to Guiyang. For example(see Lines 23–41, 91-94 in the revised manuscript):

Within the Chinese context, intended commuting functions have not been fully realized by many operational suburban railway lines, which commonly face problems of low ridership and underutilization [7-8]. For example, tourist travel has been primarily served by Beijing’s S2 Line since its opening, with commuting trips accounting for only a minor share of total ridership[9].

Concurrently, we reviewed the mainstream explanatory pathways in existing research regarding this issue and, on this basis, further identified the limitations of current studies：

To explain these phenomena, the limited commuting performance of suburban railways has been mainly attributed by previous studies to operational or institutional factors, including inappropriate station location, relatively high fares, long waiting times, and insufficient supporting infrastructure [10-12]. However, for suburban railways characterized by larger station spacing and broader service catchments, the conversion of commuting willingness into actual ridership is not only dependent on train operating services but is also strongly constrained by the built environment conditions surrounding residents’ places of dwelling. In contrast to urban metro systems, a higher sensitivity to first- and last-mile travel conditions is exhibited by commuting behavior on suburban railways. When feeder bus services are insufficient, road network connectivity is poor, or station accessibility is low, the adoption of suburban railways as a regular commuting mode may be found difficult by residents, even when latent commuting willingness exists. Consequently, persistently low utilization of such lines is often observed.

So the Guiyang Loop Line Railway—a representative example of a loop-type suburban rail system—is presented as a valuable empirical context for examining how built environment factors influence commuters’ willingness to use suburban railways.

Through these additions, we have situated the research on the Guiyang Loop Line Railway within a broader regional rail commuting framework, highlighting its theoretical value in extending the explanatory perspectives of existing literature.

Reviewers comment 4

The authors mentioned previous research on railway usage quite late into the introduction. The structure here can be considerably improved by reorganizing the logic, from broader picture to the specific case of Guiyang. It is also suggested to justify the case study as a valuable addition to the literature.

Response to comment 4

We appreciate the reviewer’s constructive comments. We agree that there was room to optimize the placement of the literature review and the overall logical flow within the Introduction. Following your suggestion, the revised manuscript has undergone a systematic restructuring of the Introduction, reorganizing the content according to the logical sequence of "Research Background—Existing Research—Research Gaps—Case Introduction—Research Objectives" (see Lines 18–31,45-98 in the revised manuscript).

In the revised version, we begin from a broader research background and systematically review the role of regional railways in supporting long-distance commuting systems within metropolitan areas, as well as the challenges currently faced in their operation:

With the continuous expansion of urban space and the deepening spatial mismatch between residential and employment locations, increasing scholarly attention has been attracted by suburban railways worldwide. The potential role of suburban rail systems in supporting metropolitan commuting structures has been highlighted by existing studies, viewed from perspectives such as urban land-use optimization, spatial structure refinement, and regional integration[5-6]. However, within the Chinese context, intended commuting functions have not been fully realized by many operational suburban railway lines, which commonly face problems of low ridership and underutilization [7-8]. For example, tourist travel has been primarily served by Beijing’s S2 Line since its opening, with commuting trips accounting for only a minor share of total ridership[9].

Subsequently, we further summarized the primary limitations of current research:

To explain these phenomena, the limited commuting performance of suburban railways has been mainly attributed by previous studies to operational or institutional factors, including inappropriate station location, relatively high fares, long waiting times, and insufficient supporting infrastructure [10-12].

However, for suburban railways characterized by larger station spacing and broader service radii, focusing solely on the supply of operational services often fails to fully reveal the underlying mechanisms behind limited commuting functionality and sparse passenger flow. Although some scholars have noted the influence of the built environment on residents' commuting behavior, the attention given to built environment factors remains significantly insufficient:

On the one hand, a station-area (destination-end) perspective has been primarily adopted in prior research, emphasizing land use patterns or development intensity around stations; conversely, systematic empirical evidence regarding the effect of residential-end built environment characteristics on commuters’ mode choice remains scarce[13]. On the other hand, most existing studies are based on large cities with relatively simple topographic conditions[14]. Mountainous cities, where urban spatial structure is strongly constrained by natural terrain, have received far less attention, and the applicability of existing findings to such contexts has yet to be adequately examined.

On this basis, we introduce the Guiyang Loop Line Railway as the research case:

Guiyang, situated in southwest China, is exemplified as a typical mountainous city characterized by steep slopes and complex terrain. As of 2025, only Metro Lines 1, 2, 3, and the S1 Line had been commissioned in Guiyang, resulting in limited coverage of the urban rail transit network. Consequently, heavy reliance is placed by residents on other public transport modes for cross-regional commuting. Simultaneously, traffic congestion has been exacerbated by factors such as insufficient road network connectivity. It is indicated in the Guiyang “14th Five-Year” Special Transportation Construction Plan that the road network structure in the central urban area remains incomplete, characterized by a low proportion of secondary and branch roads, poor network circulation, and a severe shortage of overall road network density. Consequently, further improvement is required for the traffic carrying capacity [15]. By October 2024, an increase of approximately 20,000 motor vehicles per month was recorded in Guiyang, while the urban road network density stood at only 6.7 km/km², far below the national standard of 8 km/km². This imbalance between vehicle growth and road supply has led to a pronounced disparity between vehicle proliferation and road infrastructure availability, further constraining the efficiency of cross-regional commuting[16],

To address these challenges, China’s first urban loop suburban railway—the Guiyang Loop Line Railway—was officially inaugurated in Guiyang on March 30, 2022. The line comprises a total length of approximately 113 km and includes 17 stations. Positioned as an important commuting corridor connecting multiple residential areas with the central urban area, the Loop Line Railway was designed to promote the development of the urban transport system, enhance connectivity between the city center and peripheral clusters, and support the construction of a “Guiyang–Gui’an one-hour commuting circle”. However, in comparison with its planned positioning and functional expectations, problems of low utilization and insufficient ridership have been faced by the Loop Line Railway in actual operation.

Against this background, the Guiyang Loop Line Railway—a representative example of a loop-type suburban rail system—is presented as a valuable empirical context for examining how built environment factors influence commuters’ willingness to use suburban railways. A detailed investigation into how variations in residential-side built environment characteristics shape first- and last-mile travel conditions—consequently affecting residents’ willingness to commute via the Loop Line Railway—is essential for improving operational efficiency and addressing the limitations of existing studies in terms of spatial perspective and topographic context.

Through the aforementioned adjustments, we have further strengthened the positioning of this study within the existing literature framework and emphasized its theoretical and practical significance.

Reviewers comment 5

How are the sample and survey stations determined? This requires explanation and justification. This should bring forward in the methodological section.

Response to comment 5

We appreciate the reviewer’s comments. We agree that the explanation regarding the selection of samples and survey stations in the original manuscript was not sufficiently centralized, and the methodological justification required strengthening. Following your suggestion, a new subsection titled " Sample Selection and Survey Site Determination" has been added to the Research Design section of the revised manuscript. This subsection provides supplementary details on the principles, screening process, and rationale for determining the samples and survey stations. The specific additions are as follows (see Lines 205–254 in the revised manuscript):

Sample Selection and Survey Site Determination

This study investigates residents’ willingness to commute via the Guiyang Loop Line Railway. Given that suburban railways primarily facilitate medium-to long-distance commuting between urban cores and peripheral areas, they are characterized by large inter-station spacing and significant reliance on access and egress connections. Consequently, conventional catchment area delineations based on walking distances, which are commonly applied to urban metro systems, are not directly applicable in this context. Previous studies indicate that users of suburban or commuter rail systems rely heavily on cycling, feeder buses, and other modes—in addition to walking—to complete first- and last-mile trips; as a result, acceptable access distances substantially exceed the typical walking thresholds observed for metro systems[47]. Further empirical evidence suggests that commuters’ tolerance regarding access and egress is constrained more significantly by temporal thresholds than by physical distance alone [48]. Field investigations conducted for this study indicate that, in commuting contexts, passengers generally accept access and egress travel times of 15–20 minutes, which corresponds approximately to a spatial radius of 2–3 km. In comparison to larger catchment definitions, a 3 km buffer effectively captures the primary residential clusters surrounding stations while excluding distant samples that exhibit weak relevance to suburban rail commuting decisions. Therefore, considering the functional characteristics of suburban railways, station spacing, and commuters’ access

---

## [Decision Letter · Decision Letter 1]

20 Feb 2026

Dear Dr. Zhao,

Thank you for submitting your manuscript to PLOS ONE. After careful consideration, we feel that it has merit but does not fully meet PLOS ONE’s publication criteria as it currently stands. Therefore, we invite you to submit a revised version of the manuscript that addresses the points raised during the review process.

We look forward to receiving your revised manuscript.

Kind regards,

Qing-Chang Lu

Academic Editor

PLOS One

Journal Requirements:

Additional Editor Comments:

The authors should carefully address the comments raised by the first reviewer.

Reviewer's Responses to Questions

**Comments to the Author**

Reviewer #1: All comments have been addressed

Reviewer #2: All comments have been addressed

2. Is the manuscript technically sound, and do the data support the conclusions?

Reviewer #1: Partly

Reviewer #2: Yes

3. Has the statistical analysis been performed appropriately and rigorously?

Reviewer #1: Yes

Reviewer #2: Yes

4. Have the authors made all data underlying the findings in their manuscript fully available?

Reviewer #1: No

Reviewer #2: No

5. Is the manuscript presented in an intelligible fashion and written in standard English?

Reviewer #1: No

Reviewer #2: Yes

Reviewer #1: While the authors have largely revised the introduction. Certain parts can still be improved. Specifically, the authors elaborated extensively on the case of Guiyang in terms of suburban railway development and its impact on commuting, this is still case-specific and does not really help highlight the research contribution and position the study in the literature. I would suggest move more case-specific stuff to the study context in methodology and clearly argue the contribution of the study.

The logic flow of literature review is still very unclear and requires re-organization. It taps on a suite of relevant topics including TPB, commuting, suburban railway, and in certain parts, mixed stuff. I would urge the authors to more carefully organize the topics into sub-sections with leading sentences clearly showing the logic flow across different but related topics, and last, clearly and convincingly justify the research gaps to be addressed.

In section 3.1.1, a map should be provided to map out the survey stations and possibly the distribution of the sample to allow a more straightforward presentation of the survey outcome.

Section 3.1.2, the theorical foundation can be underpinned by a conceptual framework.

Section 4.2, how was the level of significance of independent variables determined, since the XGBoost does not generate significance test?

While browsing through the results, a fundamental issue seems to be the disconnection from the behavioral theories that were introduced as the theoretical foundation. While the discussion appears to largely focus on four built environment characteristics. Due to this disconnection, the analysis and interpretations were largely simplified by overlooking the attitudinal dimensions and contradicts with the initial motivation and questions raised. This undermines the theoretical foundation and weakens the depth of analysis. I am not sure how this can be omitted in the first place.

The figures are still in poor quality.

Reviewer #2: Although the authors have provided a link to the dataset, I was unable to access it during the review process. I kindly suggest that the authors verify whether the link is available and ensure that the data can be accessed by external users. Making the dataset reliably accessible would enhance the transparency of the study.

.

Reviewer #1: No

Reviewer #2: No

---

## [Author Response · Author response to Decision Letter 2]

27 Feb 2026

Point-to-Point Responses to Reviewers

The authors would like to sincerely thank the Editor and the reviewers for their continued time and constructive comments on the revised manuscript. We greatly appreciate the valuable suggestions provided during this second round of review, which have helped us further improve the quality and clarity of the paper. We have carefully considered all comments and revised the manuscript accordingly. A detailed, point-by-point response is provided below, and all corresponding revisions have been clearly indicated in the manuscript.

Responses to comments from Reviewer # 1

Reviewers comment 1

While the authors have largely revised the introduction. Certain parts can still be improved. Specifically, the authors elaborated extensively on the case of Guiyang in terms of suburban railway development and its impact on commuting, this is still case-specific and does not really help highlight the research contribution and position the study in the literature. I would suggest move more case-specific stuff to the study context in methodology and clearly argue the contribution of the study.

Response to comment 1

Thank you for this valuable suggestion. We agree that the previous version of the Introduction contained excessive case-specific descriptions related to Guiyang, which may have weakened the clarity of the research contribution and its positioning within the broader literature. Following the reviewer’s recommendation, we have substantially revised the Introduction section. Specifically, detailed descriptions of the Guiyang suburban railway development context have been reduced and relocated to the Study Area subsection in the Research Design. The Introduction now focuses more explicitly on the research gaps in existing literature, particularly regarding the non-linear effects and spatial heterogeneity of built environment factors on suburban railway commuting behavior. In addition, we have clarified the academic contributions of this study by emphasizing three aspects: (1) integrating machine learning approaches with behavioral perspectives to analyze commuting willingness, (2) identifying threshold effects of built environment factors, and (3) revealing spatial heterogeneity in suburban railway contexts within mountainous cities. These revisions improve the theoretical positioning and highlight the novelty of the study within the existing literature. The revised text are as follows (see Lines 1–65 in manuscript)

Introduction

Suburban railways, often termed commuter rail or suburban rail, are categorized as a form of passenger rail system positioned between urban rail transit (e.g., metro systems) and intercity railways; they are characterized by commuter-oriented services, relatively high operating speeds, and large transport capacity. As a long-distance commuting mode, a crucial role is played by suburban railways in connecting central urban areas with peripheral suburbs and in supporting the development of a “one-hour commuting circle”[1]. In comparison to conventional urban rail transit, a wider spatial area is typically served by suburban railways, which feature larger station spacing, higher operating speeds, and lower service frequencies. They are primarily designed to accommodate medium-to long-distance commuting between central cities and surrounding suburban areas or satellite towns. Their functional orientation is focused on establishing efficient commuting corridors within metropolitan area[2-3]. In this context, a significant contribution is made by suburban railways toward relieving functional pressure in central urban areas, shaping rational commuting structures, facilitating convenient daily mobility, and improving accessibility in peripheral urban regions[4].

With the continuous expansion of urban space and the deepening spatial mismatch between residential and employment locations, increasing scholarly attention has been attracted by suburban railways worldwide. The potential role of suburban rail systems in supporting metropolitan commuting structures has been highlighted by existing studies, viewed from perspectives such as urban land-use optimization, spatial structure refinement, and regional integration[5-6]. However, within the Chinese context, intended commuting functions have not been fully realized by many operational suburban railway lines, which commonly face problems of low ridership and underutilization [7-8]. To explain these phenomena, the limited commuting performance of suburban railways has been mainly attributed by previous studies to operational or institutional factors, including inappropriate station location, relatively high fares, long waiting times, and insufficient supporting infrastructure [9-11].

However, for suburban railways characterized by larger station spacing and broader service catchments, the conversion of commuting willingness into actual ridership is not only dependent on train operating services but is also strongly constrained by the built environment conditions surrounding residents’ places of dwelling. In contrast to urban metro systems, a higher sensitivity to first- and last-mile travel conditions is exhibited by commuting behavior on suburban railways. When feeder bus services are insufficient, road network connectivity is poor, or station accessibility is low, the adoption of suburban railways as a regular commuting mode may be found difficult by residents, even when latent commuting willingness exists. Consequently, persistently low utilization of such lines is often observed. Nevertheless, operational and service supply factors have been the primary focus of existing studies on insufficient ridership, while limited attention has been paid to the mechanisms through which commuting willingness is influenced by the built environment, ultimately leading to low passenger volumes. On the one hand, a station-area (destination-end) perspective has been primarily adopted in prior research, emphasizing land use patterns or development intensity around stations; conversely, systematic empirical evidence regarding the effect of residential-end built environment characteristics on commuters’ mode choice remains scarce[12]. On the other hand, most existing studies are based on large cities with relatively simple topographic conditions[13], Mountainous cities, where urban spatial structure is strongly constrained by natural terrain, have received far less attention, and the applicability of existing findings to such contexts has yet to be adequately examined.

To address these research gaps, this study investigates how residential built environment characteristics influence commuters’ willingness to use suburban railways, using Guiyang, China—a typical mountainous city—as an empirical case. Specifically, this study aims to: (1) examine the nonlinear relationships between residential built environment factors and suburban rail commuting willingness; (2) identify threshold effects and spatial heterogeneity of key influencing variables using machine learning approaches; and (3) provide planning implications for improving suburban railway utilization in mountainous cities. This study contributes to the existing literature in three aspects. First, it shifts the analytical perspective from station-area characteristics to residential-side built environment factors influencing suburban rail commuting behavior. Second, it extends suburban rail research to mountainous urban contexts where terrain constraints play an important role. Third, it applies data-driven methods to reveal nonlinear and heterogeneous mechanisms underlying suburban rail use, thereby providing new insights into improving suburban railway performance.

Reviewers comment 2

The logic flow of literature review is still very unclear and requires re-organization. It taps on a suite of relevant topics including TPB, commuting, suburban railway, and in certain parts, mixed stuff. I would urge the authors to more carefully organize the topics into sub-sections with leading sentences clearly showing the logic flow across different but related topics, and last, clearly and convincingly justify the research gaps to be addressed.

Response to comment 2

Thank you very much for this insightful and constructive comment. We fully agree that a clear logical structure is essential for presenting the literature review and positioning the research contribution. Following the reviewer’s suggestion, the literature review has been substantially reorganized and improved in the revised manuscript. Specifically, the literature review is now structured into three thematic subsections with clearer logical progression:

Behavioral Theories and Commuting Intention — This subsection introduces the theoretical foundations, including the Theory of Planned Behavior (TPB) and the Technology Acceptance Model (TAM), to explain the psychological mechanisms underlying commuting intention.

Suburban Railways and Commuting Research — This subsection reviews existing studies on suburban railway systems and commuting characteristics, highlighting current research focuses and limitations, particularly the dominance of operational and supply-side explanations.

Built Environment Effects and Research Gaps — This subsection synthesizes research on built environment influences on travel behavior, identifies the limited evidence in suburban railway contexts, and clearly articulates the research gaps related to non-linear effects and spatial heterogeneity.

The research gaps are now more clearly justified, emphasizing the need to examine the non-linear and spatially heterogeneous effects of residential built environment factors on suburban railway commuting willingness. The revised text are as follows (see Lines 66–163 in manuscript)

Literature Review

Behavioral Theories and Commuting Intention

In travel behavior research, behavioral theories have been widely applied to explain individuals’ travel intentions and mode choice decisions. Among them, the Theory of Planned Behavior (TPB) and the Technology Acceptance Model (TAM) are two of the most commonly adopted theoretical frameworks. According to the Theory of Planned Behavior proposed by Ajzen, individuals’ behavioral intentions are jointly determined by attitude, subjective norms, and perceived behavioral control [14]. This framework has been extensively applied in transportation studies to explain travel mode choice behavior. Empirical evidence suggests that public transport use is significantly influenced by these three components, with attitude often identified as the most influential factor shaping travel decisions[15-17]. Other studies further highlight the critical roles of attitude and perceived behavioral control in determining individuals’ intentions to use public transportation[18-19]. The Technology Acceptance Model (TAM), originally developed by Davis[20] , emphasizes perceived usefulness and perceived ease of use as key determinants influencing individuals’ acceptance of new technologies. With the emergence of new transportation modes, TAM has increasingly been applied in transportation research, including studies on electric vehicle adoption and new energy bicycle usage[21-22]. To provide more comprehensive behavioral explanations, some studies have integrated TPB and TAM frameworks. For example, Chen combined TPB, TAM, and habit theory to analyze private car users’ willingness to shift to public transport[23]. Similarly, Wang employed an integrated TPB–TAM framework and found that perceived usefulness, perceived ease of use, attitude, subjective norms, and perceived behavioral control jointly influence individuals’ willingness to adopt sustainable travel modes[24] .These studies provide an important theoretical foundation for examining commuting willingness in emerging transportation contexts.

Suburban Railways and Commuting Research

Commuting is defined as the long-term, regular, and repetitive round-trip travel between individuals’ places of residence and employment, characterized by high regularity, persistence, and temporal rigidity[25]. With rapid economic development and continuous urban expansion, the spatial separation between workplaces and residences has increased significantly, resulting in longer commuting distances and travel times. These changes contribute to traffic congestion, reduced travel efficiency, and decreased commuter satisfaction, thereby constraining sustainable urban development[26]. Due to their high operating speed, large passenger capacity, and long service distance, suburban railways are widely regarded as critical infrastructure for supporting medium- and long-distance commuting and for facilitating “one-hour commuting circles” in metropolitan regions[27]. In Europe, North America, and East Asia, suburban rail systems were developed relatively early and have achieved relatively mature operational frameworks, as exemplified by the Paris suburban railway network[28] , the Tokyo commuter rail system[29] , and commuter rail systems in the United States[30] . Existing studies in these contexts primarily focus on commuting demand structures, passenger flow characteristics, and operational efficiency, providing valuable insights into the role of suburban railways in metropolitan commuting systems.

In contrast, suburban railways in China remain in an exploratory and developmental stage. Although several lines, such as the Shanghai R Line[31] and the Beijing S Line[32] , have been commissioned, commuting ridership performance has generally been unsatisfactory, and intended commuting functions have not been fully realized. Previous studies have largely attributed this outcome to factors related to commuter behavior and operational service attributes. For instance, limitations in route alignment, pricing, and operational flexibility were identified by Alimo as major contributors to low suburban rail ridership[11] . Irawan found that walking distance, waiting time, and parking costs significantly influence commuters’ mode choice decisions[33]. Similar conclusions have been drawn in studies conducted in China, which highlight service frequency, travel time[9] ,travel cost[34], infrastructure conditions, and development strategies[10] as key determinants of suburban rail usage.

Beyond operational attributes and fare policies, recent research has begun to emphasize the importance of commuting trip chains and access–egress conditions. Brohi, drawing on the Theory of Planned Behavior, demonstrated that commuters’ attitudes toward public transport significantly influence their willingness to use suburban railways[35]. Le further suggested that maintaining affordable fares while improving last-mile services—such as bike-sharing systems and feeder bus connections—can substantially enhance the commuting attractiveness of suburban railways[36]. Overall, although existing studies have provided a relatively comprehensive understanding of suburban railway commuting from behavioral and operational perspectives, most research remains focused on supply-side explanations or treats entire railway lines as the unit of analysis.

Built Environment Effects and Research Gaps

Against this backdrop, increasing attention has been paid to the influence of the built environment on residents’ commuting mode choice. Built environment characteristics are commonly operationalized using the well-established “5D” framework—density, diversity, destination accessibility, design, and distance to transit—to evaluate their effects on travel behavior[37]. For example, Jia Fang demonstrated that built environment attributes such as block size and intersection density are strongly associated with residents’ mode choice decisions[38]. Wang examined the driving mechanisms of urban rail transit ridership and revealed non-linear relationships and interaction effects between built environment factors and ridership, thereby providing empirical support for rail transit planning[39]. Similarly, Yafei Xi identified spatiotemporal non-linear relationships between metro station built environments and passenger flows, showing that factors such as population density and parking facility density exert significantly non-linear impacts on station-level ridership[40].

Although extensive research has confirmed that built environ

---

## [Decision Letter · Decision Letter 2]

11 Mar 2026

Dear Dr. Zhao,

Thank you for submitting your manuscript to PLOS ONE. After careful consideration, we feel that it has merit but does not fully meet PLOS ONE’s publication criteria as it currently stands. Therefore, we invite you to submit a revised version of the manuscript that addresses the points raised during the review process.

We look forward to receiving your revised manuscript.

Kind regards,

Qing-Chang Lu

Academic Editor

PLOS One

Journal Requirements:

**Additional Editor Comments:**

There are still minor revisions to be made.

Reviewers' comments:

Reviewer's Responses to Questions

**Comments to the Author**

Reviewer #1: All comments have been addressed

2. Is the manuscript technically sound, and do the data support the conclusions?

Reviewer #1: Yes

3. Has the statistical analysis been performed appropriately and rigorously?

Reviewer #1: Yes

4. Have the authors made all data underlying the findings in their manuscript fully available?

Reviewer #1: Yes

5. Is the manuscript presented in an intelligible fashion and written in standard English?

Reviewer #1: Yes

Reviewer #1: When making important statements, e.g., defining suburban railway, the unsatisfactory ridership of suburban railway, etc., it is important to cite supporting literature or document and avoid overstatements. Please carefully check the manuscript throughout.

The forms of certain citations do not align with English writing, for example, line 138: “Jia Fang demonstrated that built…” and “Yafei Xi identified spatiotemporal non-linear relationships”—We usually only use surnames when citing previous research. There are plenty other examples with similar errors, which require careful correction.

Sections 5 and 6 should be restructured. Particularly, Section 5 should discuss the key findings and actionable implications, while Section 6 should draw the conclusions and identify limitations for future works.

.

Reviewer #1: No

---

## [Author Response · Author response to Decision Letter 3]

17 Mar 2026

Responses to comments from Reviewer # 1

Reviewers comment 1

When making important statements, e.g., defining suburban railway, the unsatisfactory ridership of suburban railway, etc., it is important to cite supporting literature or document and avoid overstatements. Please carefully check the manuscript throughout.

Response to comment 1

Thank you for this valuable comment. We carefully reviewed the entire manuscript and revised statements that lacked sufficient supporting references. Specifically, additional literature has been cited when defining suburban railways and when discussing the ridership performance of suburban railway systems. Furthermore, several expressions that might be interpreted as overstatements were moderated to ensure a more cautious and evidence-based presentation. These revisions help improve the academic rigor and reliability of the manuscript. The revised text are as follows (see Lines 2-5、 170-172、178-181 in manuscript)

Suburban railways, often termed commuter rail or suburban rail, are categorized as a form of passenger rail system positioned between urban rail transit (e.g., metro systems) and intercity railways; they are characterized by commuter-oriented services, relatively high operating speeds, and large transport capacity[1].

As of October 2024, the density of the urban road network was approximately 6.7 km/km², lower than the national recommended standard of 8 km/km²[43].

The Guiyang Loop Line Railway, China’s first urban loop suburban railway, was officially opened on March 30, 2022. The line has a total length of approximately 113 km with 17 stations, forming an important transportation corridor connecting multiple residential districts and urban functional areas and supporting the development of a “one-hour commuting circle” [44].

Reviewers comment 2

The forms of certain citations do not align with English writing, for example, line 138: “Jia Fang demonstrated that built…” and “Yafei Xi identified spatiotemporal non-linear relationships”—We usually only use surnames when citing previous research. There are plenty other examples with similar errors, which require careful correction.

Response to comment 2

Thank you for this helpful comment. We have carefully reviewed the manuscript and revised the citation format to conform to standard English academic writing conventions. Specifically, only the authors’ surnames are now used when referring to previous studies in the text. Similar issues throughout the manuscript have been systematically corrected, and the manuscript has been carefully proofread to ensure consistency in citation style. The revised text are as follows (see Lines 138-139、144 in manuscript)

For example, Jia demonstrated that built environment attributes such as block size and intersection density are strongly associated with residents’ mode choice decisions[38];

Similarly, Xi identified spatiotemporal non-linear relationships between metro station built environments and passenger flows, showing that factors such as population density and parking facility density exert significantly non-linear impacts on station-level ridership[40].

Reviewers comment 3

Sections 5 and 6 should be restructured. Particularly, Section 5 should discuss the key findings and actionable implications, while Section 6 should draw the conclusions and identify limitations for future works.

Response to comment 3

Thank you for this helpful suggestion. In accordance with the reviewer’s comment, Sections 5 and 6 have been restructured to improve the organization of the manuscript. Section 5 now discusses the key findings of the study and highlights their practical implications. Section 6 summarizes the main conclusions and identifies the limitations of the study as well as directions for future research. This restructuring enhances the logical flow of the paper. The revised text are as follows (see Lines 594-712 in manuscript)

5 Disscussion and Policy Implication

Drawing on the spatial analysis of threshold effects and non-linear impacts of built environment factors, this study proposes the following recommendations to address the low ridership and underutilisation of the Guiyang Loop Line Railway, and to provide practical insights for station planning and the development of suburban railways in other cities.

(1) The results indicate that the commuting function of the Loop Line Railway depends heavily on the presence of stable and sufficiently concentrated commuting willingness. In areas with high population density and strong job - housing concentration, the Loop Line Railway is more suitable as a frequent and reliable commuting mode, and its commuter-oriented function should be prioritised. In contrast, in peripheral areas characterised by lower population density and dispersed residential patterns, changes in population scale alone are unlikely to meaningfully enhance the commuting attractiveness of the Loop Line Railway. In such contexts, improving feeder connections and reducing overall commuting time represent more effective strategies for strengthening its relative advantages.

(2) For suburban and peripheral stations, priority should be given to improving the accessibility of bus feeder services rather than expanding the overall scale of bus supply. Rather than increasing the number of bus routes, more effective measures include optimising the spatial distribution of bus stops, shortening walking distances to the nearest bus stop, and providing dedicated feeder bus services to railway stations. These measures can reduce first- and last-mile transfer costs and, in turn, enhance the commuting attractiveness of the Loop Line Railway.

(3) The model results show that the non-linear effects of road network density on commuting willingness are mainly concentrated in suburban and peripheral areas. Accordingly, road network improvements should focus on areas where transport infrastructure is relatively underdeveloped. Measures such as improving local street networks, reducing detour distances, and enhancing road connectivity can increase station accessibility and facilitate residents’ access to railway stations, thereby encouraging the use of the Loop Line Railway for commuting. In contrast, in central urban areas where road networks are already highly developed, further increases in road density may yield limited benefits. Instead, measures such as dedicated access lanes for railway transfers and enhanced traffic flow management can be adopted to reduce congestion and improve transfer efficiency.

(4) The Loop Line Railway plays an important role in mitigating the disadvantages associated with long commuting distances in peripheral areas. For residents living far from the urban core, commuting distance and time costs constitute major constraints on railway use. In these areas, the cross-regional rapid accessibility provided by the Loop Line Railway is more likely to be strongly perceived and valued. Accordingly, in the process of optimising urban spatial structure, the Loop Line Railway can serve as an important instrument for improving commuting conditions in peripheral areas and promoting shifts in commuting modes. However, given the relatively low level of development in these areas, particular attention should be paid to strengthening feeder transport facilities and reducing overall commuting time. From a behavioral perspective, optimizing built environment conditions around stations not only improves objective accessibility but also enhances perceived usefulness and perceived behavioral control, thereby strengthening residents’ willingness to use suburban railways for commuting.

6 Conclusions and Limitations

6.1 Conclusions

Based on survey data on residents’ willingness to commute via the Guiyang Loop Line Railway, this study systematically examines the non-linear effects of built environment factors on commuter willingness and their spatial heterogeneity. The main conclusions are as follows:

(1) Population density, distance to the nearest bus stop, road network density, and distance from the city center represent the four built environment factors with the most substantial impact on commuting willingness. Specifically, population density and road network density exert a positive influence, whereas distance to the nearest bus stop and distance from the city center show an inverse correlation with commuting propensity for the Loop Line Railway.

(2) The non-linear dependence of commuting willingness on the built environment exhibits strong spatial non-stationarity along the Loop Line Railway. The significance of threshold effects and the intensity of non-linear impacts associated with different built environment factors vary substantially across space, forming a clear gradient from the urban core to suburban and peripheral areas. Compared with the central urban area, residents living around suburban and peripheral stations show greater sensitivity to changes in feeder bus conditions, road network connectivity, and commuting distance, which results in more pronounced threshold effects and stronger non-linear responses.

(3) The mechanisms through which built environment factors influence commuting willingness differ across spatial contexts, with peripheral areas showing greater potential for improving commuting willingness. Specifically, the threshold effects and non-linear impacts of population density are primarily concentrated around central urban stations, indicating that in areas with highly concentrated commuting willingness, changes in population scale have stronger marginal effects on willingness to commute via the Loop Line Railway. In contrast, the non-linear effects of distance to the nearest bus stop and road network density are more pronounced around suburban and peripheral stations, indicating that in areas with relatively underdeveloped public transport and road infrastructure, first- and last-mile connection conditions and station accessibility represent key constraints on using the Loop Line Railway for commuting. In addition, the non-linear effect of distance to the city center is most pronounced at stations located farther from the urban core, suggesting that the Loop Line Railway has greater potential to mitigate the spatial disadvantage associated with long commuting distances in peripheral areas.

(4)Furthermore, the findings of this study also provide empirical support for the behavioral mechanisms proposed by the Theory of Planned Behavior (TPB) and the Technology Acceptance Model (TAM). The results indicate that built environment characteristics do not influence commuting willingness solely through physical accessibility constraints, but also indirectly by shaping residents’ perceptions of convenience, usefulness, and travel effort. Improvements in population density, feeder accessibility, and road connectivity may enhance perceived usefulness, perceived ease of use, and perceived behavioral control associated with the Loop Line Railway, thereby strengthening commuting intention. Conversely, unfavorable spatial conditions may reduce perceived advantages and increase psychological resistance to rail commuting. These findings highlight the importance of integrating objective spatial planning with perception-oriented service improvements to promote suburban railway utilization.

6.2 Limitations and Future Research Directions

Although this study provides a systematic analysis of the non-linear effects and spatial heterogeneity of built environment factors on commuting willingness, several limitations remain that warrant further investigation.

First, this research is limited to the Guiyang Loop Line Railway as a single case studay. As a typical mountainous city, Guiyang possesses distinctive topographical features, urban spatial structures, and a unique rail transit development stage. Therefore, caution is required when generalizing the findings to cities with divergent geographical conditions or development trajectories. Future studies should incorporate comparative analyses across multiple cities or railway lines to validate the robustness and generalizability of the conclusions.

Second, the analysis relies on cross-sectional survey data, which captures residents’ stated commuting willingness at a single temporal point. Consequently, the study cannot fully capture dynamic shifts in commuting willingness resulting from built environment improvements or operational adjustments. Future research could benefit from integrating longitudinal survey data with objective travel data (e.g., smart card or mobile phone signaling data) to better examine the long-term behavioral impacts of built environment interventions.

Finally, data constraints limited the built environment measurement to indicators such as population density, road network density, and bus stop proximity. Other potentially relevant factors, such as pedestrian environment quality and public transport reliability, were not explicitly considered. Future studies should incorporate finer-grained spatial and environmental data to elucidate the mechanisms through which the built environment influences suburban railway commuting behavior.

Responses to comments from Editor

Reviewers comment 1

Response to comment 1

Thank you for this reminder. We have carefully reviewed the reference list to ensure that all references are complete, accurate, and relevant to the study. A small number of additional references have been incorporated where appropriate to strengthen the literature support of the manuscript. We also confirmed that none of the cited references have been retracted.

---

## [Editor Report · Decision Letter 3]

18 Mar 2026

Spatial heterogeneity of the nonlinear impact of built environment on suburban railway commuting willingness: Taking Guiyang Loop Line Railway as an example

PONE-D-25-62306R3

Dear Dr. Zhao,

We’re pleased to inform you that your manuscript has been judged scientifically suitable for publication and will be formally accepted for publication once it meets all outstanding technical requirements.

Kind regards,

Qing-Chang Lu

Academic Editor

PLOS One
---

## [Editor Report · Acceptance letter]

PONE-D-25-62306R3

PLOS One

Dear Dr. Zhao,

I'm pleased to inform you that your manuscript has been deemed suitable for publication in PLOS One. Congratulations! Your manuscript is now being handed over to our production team.

Kind regards,

on behalf of

Dr. Qing-Chang Lu

Academic Editor

PLOS One